# HOT PATE:
# PRIVATE AGGREGATION OF DISTRIBUTIONS FOR DIVERSE TASKS

## ABSTRACT

The Private Aggregation of Teacher Ensembles (PATE) framework is a versatile approach to privacy-preserving machine learning. In PATE, responses made based on different parts of sensitive data are aggregated into a single response in a privacy-preserving way. Recently, multiple works applied PATE for tasks such as sequential text generation that are inherently diverse (or "hot"), with multiple valid responses. These designs, however, suffer from tension between diversity and privacy – since diversity in the responses reduces agreement which forces the aggregation to use smaller noise scales and thus incur higher privacy loss. But limiting diversity of the aggregate response is undesirable since in modern large language models, the very knowledge we want to transfer is encapsulated in the response distribution. We propose *hot PATE* that is tailored for the diverse setting where responses are distributions. We formally define *preserving diversity* and design an efficient aggregation method that provably transfers the diversity to the (randomized) aggregate response while incurring no privacy penalty. The method can be implemented using an API access to proprietary models and used as a plug-in replacement for the baseline "cold" PATE in existing tools. We demonstrate empirically the potential of hot PATE for an order of magnitude improvement in a task of in-context learning via prompts.

## 1 INTRODUCTION

Generative AI models, such as large language models (LLMs), are incredibly powerful tools that can be fine-tuned for specific contexts, even without explicit supervision (Radford et al., 2019; Brown et al., 2020). Generative models diverge from conventional machine learning models in that they support open ended, *diverse* tasks, where there are multiple appropriate responses, and this very flexibility is essential for much of their functionality. Diversity is typically tuned via a temperature parameter in the softmax, with higher temperature yielding higher entropy (more diverse responses). Furthermore, when evaluating the coverage or extracting knowledge from a trained model, such as for distillation tasks, the conventional approach involves querying the model on a prepared (sampled or curated) test set of examples. However, with generative AI models, the knowledge coverage on a specific domain is often encapsulated by the output distribution itself to a general instruction as part of a *prompt* to the model, and can be evaluated or retrieved by sampling this distribution.

Frequently there is a need to train models or fine-tune publicly-available foundation models using sensitive data such as medical records, incident reports, or email messages. In this case, privacy must be preserved in the process. Specifically, we consider the strong mathematical guarantees of differential privacy (DP) (Dwork et al., 2006). An approach that achieves privacy by modifying the training process is DPSGD (Abadi et al., 2016), where noise is added to clipped gradient updates. DPSGD can also be applied with fine tuning (Yu et al., 2022; Duan et al., 2023; Kurakin et al., 2024). An alternative approach, that only relies on black box training and use of models that are not privacy-preserving, is the Private Aggregation of Teacher Ensembles (PATE) paradigm (Papernot et al., 2017; Bassily et al., 2018; Papernot et al., 2018). PATE follows the "sample and aggregate" method of Nissim et al. (2007). We describe the basic workflow which we refer to here as *cold* PATE on how the ensemble is used to label a set of new examples $X$ while protecting the privacy of the training data:

---

**Cold PATE**

1. Partition the sensitive dataset $D$ into $n$ parts $D = D_1 \sqcup \cdots \sqcup D_n$. For $i \in [n]$, train a *teacher* model $M_i$ on data $D_i$.

2. Repeat the following:

   - Input an example $x \in X$.
   - For each teacher $i \in [n]$, apply $M_i$ to $x$ and obtain a label $y_i := M_i(x) \in V$.
   - Compute the frequency histogram $\boldsymbol{c}$:

   $$\text{for } j \in V, \; c_j = \sum_{i \in [n]} \mathbb{1}\{y_i = j\}. \tag{1}$$

   - DP aggregate the histogram $\boldsymbol{c} \mapsto y$ to obtain a single label $y \in V$ (or abort if there is insufficient agreement). Output $y$.

---

Differential privacy requires that the output distribution is stable to a change of a single data record. In the PATE framework, the votes histogram of each example is stable to a change of one record in $D$: At most one teacher, the one trained on this record, is affected and thus may change its vote. Therefore, at most two frequency counts $c_j$ may change in the histogram, and each by at most 1. A noisy selection of a label from the histogram, that hides this small difference in the counts, is therefore privacy preserving.

The labels may be the end goal or the set of privacy-preserving labeled examples $\{(x, y)\}$ can be used to train a student model. The limitations of cold PATE are that it was originally designed for classification-like tasks, where each example $x$ has a single ground-truth label $y \in V$. Moreover, there is a need for a source of unlabeled non-private training examples to facilitate the knowledge transfer to the student. This is unsatisfactory because generative AI models support tasks with responses that are diverse and open ended. Moreover, knowledge is encapsulated in the diversity of the response distribution and there is a promise of transferring knowledge to the student in a more fluid way. We thus ask the following question:

> *Can we design a version of PATE that is effective for diverse and open-ended tasks*
> *and unleashes more of the capabilities of generative models?*

**Application for in-context learning** One motivation for our study is the effectiveness of in-context learning via *prompts*. A prompt is an engineered prefix with a task that is given to the base model. Prompts can include specific instructions and/or a set of *shots* (scenario exemplars). Prompts are appealing for multiple reasons: A small number of shots (Liu et al., 2021) often outperform tailored trained models (Zhou et al., 2022; Garg et al., 2023). Prompting is efficient, as it is simply inference – there is no need for parameter updates. Finally, prompts only requires API access to the model, which is important given the trend towards proprietary models.

When our data is sensitive, we would like the end product to be privacy-preserving. Concretely, consider the task of generating a representative set of synthetic privacy-preserving data records from a set of sensitive data records. The sensitive records may include components that are identifying and components that are shared with many other records. A privacy-preserving aggregation ensures that the synthetic records do not include identifying information. Additionally, it is essential to *preserve diversity* in order to ensure coverage, that is, that our set of synthetic records is indeed representative of the sensitive records. The synthetic records that are generated can then be used to train a student model that is not necessarily generative, fine-tune a generative model (OpenAI, 2023), or construct a privacy-preserving student prompt for downstream tasks. The latter allows for harnessing the ability of generative models to generalize from few shots.

We seek a PATE mechanism that supports the following. Each teacher is assigned a disjoint subset of sensitive data records. These data records are used to construct a prompt that also includes an instruction of the form "generate a representative data record given this example set of data records." Each teacher then has its own distribution on responses. By repeating multiple times we can obtain different samples that are a representative set of shots. We then hope to aggregate responses of different teachers in a way that preserves both diversity and privacy. This design is appealing as there is little cost to scaling up the number of teachers: Each teacher is simply a prompted base model and there is no need for training or significant storage. Prompts are inexpensive, the current OpenAI API

supports $10^5$ context/output tokens for US$5-$10 (OpenAI, 2023a). The bottleneck to scaling up the number of teachers is thus the amount of available sensitive data. Scaling up is highly beneficial because generally with DP aggregation, the number of queries we can support for a given privacy budget is quadratic in the number of teachers.

**Diversity-privacy tradeoff:** An issue that arises when applying cold PATE with high diversity is that utility rapidly deteriorates with diversity. To see this, assume there are $r$ good responses with equal probabilities. Note that higher $r$ means more diversity. The $n$ teachers votes would then be split with $\approx n/r$ teacher votes per option. This lower agreement means that in order to return any of the answers we must use privacy noise of scale $\sigma < n/r$. This inverse dependence of noise with $r$ means the privacy loss must increase with $r$. We can attempt to remedy this via some tie-breaking (e.g., each teacher selects a response in the top-$k$ with the largest index). This does result in high agreement but we lose the diversity in the output that is needed to facilitate a fluid knowledge transfer. All prior and concurrent works we are aware of for privacy-preserving sequential text generation or in-context learning via prompts (Tian et al., 2022; Duan et al., 2023; Wu et al., 2023) either ignored this issue or addressed it by reducing or limiting diversity (see discussion in Section A). We ask the following:

*Is the diversity-privacy tradeoff indeed inherent?*

OVERVIEW OF CONTRIBUTIONS AND ROADMAP

We propose *hot PATE*, described in Section 2. The method is suitable for auto-regressive models and diverse and open ended tasks, where the appropriate response is a sample from a distribution. With hot PATE, each teacher $i \in [n]$ at each step computes a "next token" distribution $\boldsymbol{p}^{(i)}$ over tokens $V$. These distributions are aggregated so that the response token from the ensemble is sampled from that aggregate distribution. The heart of our design is an aggregation method that preserves privacy and critically also the diversity of the teachers distributions. Our primary technical contributions are mathematically formalizing this requirement and proposing aggregation methods where there is *no penalty* with increased diversity. Hot Pate can be added in a black-box manner to existing designs for in-context learning via prompts to improve the utility privacy tradeoff.

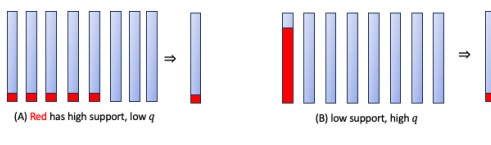

(A) Red has high support, low $q$    (B) low support, high $q$

Figure 1: Illustration of two sets of probability distributions, each shown as a rectangle with the red portion representing the probability of token $j$. The left set corresponds to high teachers' support for low probability $q$. The right set to low teachers' support for high $q$. The probability of token $j$ in the average distribution is the same in both cases.

In Section 3 we motivate and formalize a definition of robustly *preserving diversity*, which allows for knowledge transfer that is compatible with limitations imposed by privacy. A natural diversity-preserving approach is for each teacher $i \in [n]$ to contribute a token $y_i$ sampled independently from $\boldsymbol{p}^{(i)}$. We refer to this as *independent ensemble*. The resulting vote histogram is what is produced by cold PATE (Papernot et al., 2017; 2018; Duan et al., 2023) when applied in a diverse setting. The histogram can then be DP aggregated to produce a response token. The privacy loss depends on the frequency (count) of the response token. With independent samples, this count is concentrated around the *average* probability of the token across teachers. This probability is smaller when there is high diversity. Therefore, independent ensembles as an intermediate step inherently result in privacy guarantees that sharply deteriorate with the diversity of teacher distributions. We argue that this higher privacy noise may or may not be necessary, and this depends on properties of the teacher distributions that are lost by independent ensembles. The frequency histograms produced by independent ensembles are concentrated around the average of the teachers' distributions. The issue, as depicted in Figure 1, is that averaging loses a critical distinction between high teachers' support with low probability $q$ (which we can hope to transfer in a privacy-preserving manner) and low support with high $q$ (which can not be transferred in a privacy-preserving manner). Our definition of robust diversity transfer makes this important refinement: A token is required to be transferred to the aggregate only when there is sufficient teachers' support. Informally, for a robustness parameter $\tau \in [n]$, there are two requirements:

- (transfer requirement) Any token that has probability at least $q > 0$ (no matter how small) across $c$ teachers where $c \geq \tau$, is "transferred" in that it has probability $\Omega(qc/n)$ in the aggregate distribution.
- (relevance requirement) We do not transfer irrelevant tokens, that is, for any token $j$, its probability in the aggregate distribution is not much higher than its average probability in the teacher distributions.

As argued, independent ensembles lose the robustness signal. In Section 4 we propose the method of *ensemble coordination*. A coordinated ensemble samples a shared randomness and based on that, each teacher $i$ contributes a token $y_i$. The marginal distribution of each $y_i$ is $\boldsymbol{p}^{(i)}$, same as with independent ensemble. But the difference is that teachers votes are maximally positively correlated. The frequency $c_j$ of token $j$ has high spread and in particular can (roughly) be $\Omega(\tau)$ with probability $\Omega(q)$. This property facilitates DP aggregation with no penalty for diversity. With coordinated ensembles, two teachers with very diverse distributions that have a small total variation distance produce the same token with probability that depends on the distance. In particular, when the distributions are equal (the distance is 0), the same token would be produced.

In Section 5 we empirically demonstrate the properties and benefits of ensemble coordination for a simple task of in-context learning via prompts on the Llama 3 language model (lla, 2024). We evaluate the coverage and diversity of aggregate distributions formed by only transferring frequency counts that exceed a threshold $T$. We observe an order of magnitude improvement over the baseline of independent ensembles in terms of the value of $T$ needed to achieve a certain coverage and in terms of diversity of the aggregate. Recall that larger $T$ means that we can use more noise (noise scale is proportional to $T$) and thus incur lower privacy loss.

DP aggregation methods for histograms that apply with independent ensembles, such as Papernot et al. (2017; 2018), can be applied in an off-the-shelf manner with histograms generated by coordinated ensembles. The primary gain of hot PATE is in the utility privacy tradeoff. In Section D we present DP aggregation schemes that are applied to frequency histograms generated by coordinated ensembles and return a token. We establish that the end-to-end process preserves diversity in the sense that it satisfies our formal requirements (Section 3). We distinguish between two application scenarios of applications with *homogeneous* or *heterogeneous* ensembles (see Figure 2). Homogeneous ensembles are formed by randomly partitioning a sufficient number of data records among teachers. The assumption then is the same as with cold PATE – most teachers possess the core knowledge we wish to transfer (see Figure 3 (A)). In this case it suffices to require diversity preservation with large support $\tau = \Omega(n)$ and the aggregate we need is simply a (noisy) *maximizer* of the histogram. Heterogeneous ensembles may arise when each teacher is an agent of one or few users. In this case, we want to preserve diversity both within and across teachers and for the latter it is necessary to allow smaller groups of teachers to support each transfer, that is, set a smaller $\tau$ (see Figure 3 (B)). In this case, a diversity-preserving aggregate is a *weighted sample* from the histogram.

In Sections E and F we further explore privacy analysis methods that are data dependent and can increase the number of queries processed for a given privacy budget by orders of magnitude. In particular, for token-by-token sequential text generation there are many steps and the cost of naive DP composition is prohibitive. What makes the approach feasible is that many of the steps have high agreement (similar teacher distributions where coordinated ensembles generate high agreement histograms). With data dependent analysis, steps with high agreement (or no agreement) can be essentially free. Moreover, with heterogeneous ensembles we can charge teachers individually (instead of the whole ensemble) and only for steps in which the teacher contributed to the final token (Hassidim et al., 2020; Cohen and Lyu, 2023).

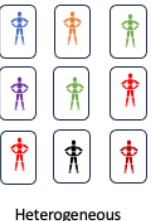
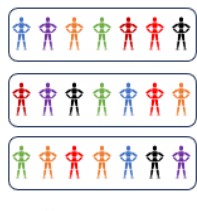

Heterogeneous          Homogeneous

Figure 2: Ensemble types for Hot Pate. In homogeneous ensembles each teacher gets a representative part of the data. In heterogeneous ensembles each teacher has the data of one or few users (aka "privacy units").

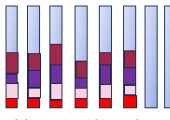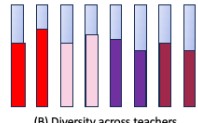

(A) Diversity within teachers    (B) Diversity across teachers

Figure 3: Illustrating diversity *within* teachers, that stems from semantic similarity or knowledge encapsulated in the base model or few exemplars. In this case, coordinated ensembles form high agreement and a higher $\tau$ suffices. Diversity *across* teachers, stems from data that is available only to few teachers. Coordinated ensembles reflect it and require lower $\tau$.

## 2 PATE FOR SEQUENTIAL TEXT GENERATION

We use the term *tokens* for elements of the input and response strings. We denote the vocabulary of tokens by $V$. For an input context (prompt) $T \in V^*$, a response sequence $R$ is generated sequentially token by token. Specifically, the next token at each step, is sampled from a probability distribution over $V$ that depends on the current context (concatenation of the prompt and response prefix) $T \cdot R$. The probabilities are computed from weights (logits) $(w_j)_{j \in V}$ produced at inference by the model and a *temperature* parameter $t > 0$, using a softmax function:

$$p_j := \frac{e^{w_j/t}}{\sum_{i \in V} e^{w_i/t}} \,.$$

In low temperatures, the highest weight token $\arg\max_j w_j$ has probability close to 1. As we increase the temperature, the probability distribution flattens with similarly-weighted tokens having similar probabilities. *Cold* temperature is appropriate for classification-like tasks with one correct response and *hot* temperature is appropriate for diverse tasks. We therefore refer to the basic PATE as *cold* PATE and to our proposed method that is tailored for diversity as *hot* PATE.

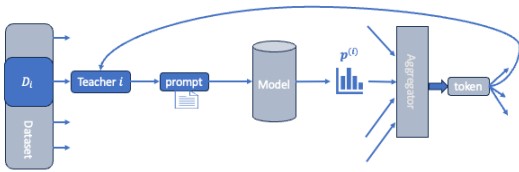

Figure 4: Sequential text generation with diversity

PATE for sequential text generation is illustrated in Figure 4. The data $D$ is partitioned to disjoint parts $D_i$ ($i \in [n]$). A prompt $T_i$ is constructed from data part $D_i$. We then generate a sanitized response sequence $R$ of tokens. We initialize $R \leftarrow \{\}$ and proceed sequentially in lockstep, by repeating the following:

1. For $i \in [n]$: Let $\boldsymbol{p}^{(i)}$ be the output distribution over $V$ when querying the model with the prompt $T_i$<instruction to complete prefix>$R$.

2. Apply a privacy-and-diversity preserving randomized aggregation $\mathcal{M}((\boldsymbol{p}^{(i)})_{i \in [n]}) \mapsto y$, where $y \in V$.

3. Concatenate $R \leftarrow R \cdot y$.

This open-ended design can be used with an instruction to generate a student prompt or representative synthetic shots. This aligns with the demonstrated and evolving capabilities of contemporary large language models and prompt engineering. Such instructions may generate diverse responses and the objective is that what is *transferred*, which is captured by the aggregate distribution $\mathcal{M}((\boldsymbol{p}^{(i)})_{i \in [n]})$, preserves the diversity present in the teacher distributions $(\boldsymbol{p}^{(i)})_{i \in [n]}$. The main difference between the baseline cold PATE and our proposed hot PATE is in the aggregation $\mathcal{M}$ in step (2). We first describe the aggregation with cold PATE (Duan et al., 2023) and present our aggregation mechanism for hot PATE in subsequent sections.

## 2.1 COLD PATE: INDEPENDENT ENSEMBLE

Each teacher $i \in [n]$ samples *independently* $y_i \sim \boldsymbol{p}^{(i)}$. The frequency histogram $(c_j)_{j \in V}$ is computed as in (1).[1] The DP aggregation mechanism adds noise to each $c_j$ to obtain a privacy-preserving sanitized histogram $(\tilde{c}_j)_{j \in V}$. We then select a token. A baseline meta-method, `NoisyArgMax` select the maximizer $\arg\max_j \tilde{c}_j$ (Duan et al., 2023). The privacy cost of this aggregation inversely depends on the noise scale $\sigma$,[2] which for utility, must satisfy $\sigma \ll \max_j c_j$.

When the distributions are more diverse, $\max_j c_j$ is smaller so for utility we must use a smaller $\sigma$. Moreover, $\arg\max_j \tilde{c}_j$ is not diversity preserving: If the most frequent token is $j$ and we have a token $h$ with frequency $c_h = c_j/2$, we still want to select $h$ with probability that is $1/2$ of that of token $j$, that is, select a weighted sample from the histogram. To do this in a privacy-preserving way we must use an even smaller noise scale that depends on the smallest counts that we aim to transfer.

## 3 DIVERSITY-PRESERVING AGGREGATION

Diversity and privacy appear to be conflicting in that DP in its essence requires that the output token is supported by sufficiently many teachers. But to preserve diversity we need to also transfer tokens that have low probability in the teacher distributions to the aggregate distribution. The most natural candidate for an aggregate distribution that preserves diversity is the average teacher distribution $\frac{1}{n} \sum_{i \in [n]} \boldsymbol{p}^{(i)}$, which is essentially what independent ensembles use. The caveat is the issue pointed out in the introduction (see Figure 1): It does not distinguish between tokens that are in the support of the distributions of very few teachers with high probability and those that are in the support of many teachers, with low probability. The privacy loss with independent ensembles (cold PATE) depends, in both cases, on the lowest average values we wish transferred. We propose a more nuanced requirement of preserving diversity that makes this distinction and is parametrized by a robustness parameter $\tau$, that corresponds to the number of supporting teachers. We then propose privacy preserving mechanisms that preserve diversity with privacy loss that depends only on $\tau$, regardless of how diverse the teacher distributions are.

**Definition 1** (Diversity-preserving aggregation of distributions). Let $f(\boldsymbol{p}^{(i)})_{i \in [n]}) \mapsto \boldsymbol{P}$ map from $n$ probability distributions over $V$ to a probability distribution over $V \cup \{\perp\}$. We say that $f$ is *diversity-preserving* with $\tau \in \mathbb{N}$, $\beta \in (0,1]$, $\gamma \geq 1$ if for any input and $j \in V$

1. For all $q \in [0,1]$,

$$\left(c_{j,q} := \sum_{i \in n} \mathbb{1}\{p_j^{(i)} \geq q\}\right) \geq \tau \implies P_j \geq \beta \cdot \frac{c_{j,q}}{n} q \ .$$

2. $P_j \leq \gamma \frac{1}{n} \sum_{i \in [n]} p_j^{(i)}$ .

The first property is that probability $q$ across enough ($\tau$) teachers, no matter how small is $q$, is transferred to the aggregate distribution. The second ensures that we do not output irrelevant tokens.

Requirements are stricter (and can be harder to satisfy) when $\beta$ and $\gamma$ are closer to 1 and when $\tau$ is smaller. A setting of $\tau = 1$ and $\beta = \gamma = 1$ allows only for the average distribution to be the aggregate. A larger $\tau$ increases robustness in that more teachers must support the transfer.

**Remark 1** (failures). *It is necessary to allow for $\perp$ (failure) in the support of the aggregate distribution when $\tau > 1$. For example, when the prompt instruction ask for a patient ID, and assuming no generalization, the teacher distributions have disjoint supports and no token can be returned. Failures in the generation can be addressed by: (i) Repeating the step with different shared randomness (ii) sample a token from a non-private default prompt or model, or (iii) redesign the prompt instruction.*

---

[1] Alternatively, instead of sampling, we can use the expected values $\bar{c}_j := \sum_i p_j^{(i)}$. The values $\bar{c}_j$ are a scaled by $n$ average of teacher distributions. The histogram $((\bar{c}_j)$ has the same privacy properties as an independent sampled histogram $(\bar{c}_j)$, since the impact of a data point on the $\ell_1$ norm is bounded by 1. Additionally, with independent ensemble, $c_j$ is anyhow concentrated around $\bar{c}_j$ so the respective noisy counts are close $\tilde{\bar{c}}_j \approx \tilde{c}_j$.

[2] Our discussion applies to all mechanisms of this form, see review in Section D of particular noise distributions.

---

**Algorithm 1:** `CoordinatedSamples`

---

**Input:** Teacher distributions $(\boldsymbol{p}^{(i)})_{i \in [n]}$

**foreach** *token* $j \in V$ **do** sample i.i.d. $u_j \sim \mathsf{Exp}[1]$  // Sample shared randomness $\rho = (u_j)_{j \in V}$

**foreach** *teacher* $i$ **do**  // Compute coordinated samples $(y_i)_{i \in [n]}$

$\quad \lfloor \ y_i \leftarrow \arg\max_j \frac{p_j^{(i)}}{u_j}$  // bottom-$k$ sampling transform

**foreach** *token* $j \in V$ **do**  // Compute frequencies

$\quad \lfloor \ c_j \leftarrow \sum_{i \in [n]} \mathbb{1}\{y_i = j\}$

**return** $(c_j)_{j \in V}$, $\rho = (u_j)_j$  // Histogram of frequencies

---

**Remark 2** (Setting of $\tau$). Homogeneous ensembles *occur when data is randomly partitioned so that most teachers receive a representative part and possess the knowledge we wish to transfer. The goal is to transfer the parts of the distributions that are common to most teachers and $\tau > n/2$ suffices. In* heterogeneous ensembles*, each teacher might have data from one or very few "users." This arises when each teacher has small capacity (prompts currently have limited size of 8k-64k tokens (OpenAI, 2023b)) or when by design each teacher is an agent of a single user. The goal here is to transfer parts of the distribution that are common to smaller subgroups of teachers and set $\tau \ll n$.*

## 4 ENSEMBLE COORDINATION

We propose *ensemble coordination* and establish that it facilitates privacy and diversity preserving aggregation. As with independent ensembles, for $n$ probability distributions over $V$ the ensemble produces a histogram $(c_j)_{j \in V}$ over $V$ with total count $\sum_{j \in V} c_j = n$. The sampling of $\boldsymbol{c}$ by a coordinated ensemble is described in Algorithm 1. The algorithm samples shared randomness $\rho := (u_j)_{j \in V}$. Each teacher $i \in [n]$ then contributes a single token $y_i \in V$ that is a function of its distribution $\boldsymbol{p}^{(i)}$ and $\rho$. The frequencies $c_j$ are computed as in (1).

The sampling method in ensemble coordination is a classic technique called *coordinated sampling*. It was first introduced in statistics works in order to obtain samples that are stable under distribution shifts (Kish and Scott, 1971; Brewer et al., 1972; Saavedra, 1995; Rosén, 1997; Ohlsson, 2000) and in computer science works for computational efficiency via sampling-based sketches and a form of Locality Sensitive Hashing (LSH) (Cohen, 1994; 1997; Broder, 2000; Indyk and Motwani, 1998; Haas, 2011). Its recent applications include private learning (Ghazi et al., 2021) and speculative decoding (Leviathan et al., 2023).

### 4.1 PROPERTIES OF COORDINATED HISTOGRAMS

Let $(\boldsymbol{p}^{(i)})_{i \in [n]}$ be probability distributions over $V$ and let $Y_{\mathrm{coo}}$ and $Y_{\mathrm{ind}}$ be the respective distributions of votes $(y_i)_{i \in [n]}$ generated by a coordinated or independent ensemble with teacher distributions $(\boldsymbol{p}^{(i)})_{i \in [n]}$. Let $H(Y_{\mathrm{coo}})$ and $H(Y_{\mathrm{ind}})$ be the respective distributions of histograms.

For each token $j$, its expected frequency, over the sampling of histograms, is the same for coordinated and independent ensembles:

**Claim 1** (Expected token frequency).

$$\forall j \in V, \ \mathsf{E}_{\boldsymbol{c} \sim H(Y_{\mathrm{coo}})}[c_j] = \mathsf{E}_{\boldsymbol{c} \sim H(Y_{\mathrm{ind}})}[c_j] = \sum_i p_j^{(i)} \ . \tag{2}$$

*Proof.* The marginal distribution of $y_i$ for teacher $i$ is $\boldsymbol{p}^{(i)}$ with both independent and coordinated ensembles and thus the claim follows from linearity of expectation. $\qquad \square$

In a coordinated ensemble, votes of different teachers are much more likely to agree than in an independent ensemble (see Section B for a proof):

**Claim 2** (Agreement probability). *For different teachers $i, k \in [n]$ and token $j \in V$, the probability that both samples agree on token $j$ is*

$$\Pr_{\boldsymbol{y} \sim Y_{\mathrm{coo}}} [y_i = y_k = j] = \frac{\min\{p_j^{(i)}, p_j^{(k)}\}}{\sum_j \max\{p_j^{(i)}, p_j^{(k)}\}} \in \left[\frac{1}{2}, 1\right] \cdot \min\{p_j^{(i)}, p_j^{(k)}\}$$

$$\Pr_{\boldsymbol{y} \sim Y_{\mathrm{coo}}} [y_i = y_k = j] \geq \Pr_{\boldsymbol{y} \sim Y_{\mathrm{ind}}} [y_i = y_k = j] = p_j^{(i)} \cdot p_j^{(k)} \ ,$$

*with equality possible only when $\max\{p_j^{(i)}, p_j^{(k)}\} = 1$.*

The key feature of coordinated histograms is that we can generate a sample from a diversity-preserving aggregate distribution as in Definition 1 by exclusively considering tokens that appear with frequency at least $\tau/2$ in the histogram (see Section B for a proof):

**Theorem 1** (Utility of Coordinated Ensembles). *We can sample from an aggregate distribution that satisfies Definition 1 with parameters $\tau$, $\beta = 0.34$ and $\gamma = 2$ by sampling a coordinated histogram $\boldsymbol{c} \sim H(Y_{\mathrm{coo}})$ and only considering tokens $j$ with $c_j \geq \tau/2$.*

### 4.2 PRIVACY PROPERTIES

With both independent and coordinated ensembles, we aggregate the histogram in a privacy-preserving way to obtain one token. The distribution of the histograms produced by these ensemble types is very different. But the privacy properties in terms of the divergence between neighboring datasets are identical and immediate:

**Observation 1.** *For every fixture of the shared randomness $\rho$, changing one of the distributions $\boldsymbol{p}^{(i)}$ given as input to Algorithm 1 changes at most one item of the resulting histogram. That is, letting $H$ and $H'$ denote the resulting histograms before and after the modification, we have that $H, H'$ are at Hamming distance 2 (viewed as vectors in $\mathbb{N}^{|V|}$).*

The following corollary is immediate from Observation 1.

**Corollary 1.** *Let $\mathcal{A}$ be an algorithm whose input is a histogram $H \in \mathbb{N}^{|V|}$, such that for any two neighboring histograms $H, H'$ (differing by at most one item) it holds that $\mathcal{A}(H) \approx_{(\varepsilon, \delta)} \mathcal{A}(H')$. Then the composed algorithm $\mathcal{A}(\mathtt{CoordinatedSamples}(\cdot))$ is $(\varepsilon, \delta)$-differentially private.*[3]

Therefore, we can apply off-the-shelf the same DP aggregation schemes we would use with independent ensembles (see Section 2.1) to coordinated ensembles. The benefit of coordinated ensembles, per Theorem 1, is a much more favorable utility privacy trade-off: It suffices to set the privacy noise scale to $\propto \tau$ *regardless of diversity*, whereas with independent ensembles we must scale the noise down with diversity to obtain utility. This benefit broadly holds with any DP histogram sanitizing method[4] that selects from tokens with count $\tau/2$ whereas tokens with low or zero counts are filtered out, including the `NoisyArgMax` methods used with cold PATE (Papernot et al., 2017; 2018). As mentioned in Section 2.1, the selection of a token from a sanitized histogram can be `NoisyArgMax` for homogeneous ensembles or a weighted sample for heterogeneous ensembles (Remark 2) – see details in Section D. Simulation results with a particular $(\varepsilon, \delta)$-DP analysis method are reported in Section E.

### 4.3 IMPLEMENTATION IN LANGUAGE MODELS

Coordination can be implemented preferably with, but also without, an enhanced API access to a proprietary base model: (i) Model-side: the shared randomness $\rho$ is provided as input along with each query prompt and the response token is sampled using $\rho$. (ii) Application-side: API returns the full distribution and sampling done in the application (iii) No API enhancements: we can approximate the distribution by repeated sampling with the same prompt. This impacts computation since the number of samples required increases with diversity but does not impact privacy. Our demonstration in Section 5 with a public model (AI@Meta, 2024) uses the full distribution.

---

[3]This corollary holds for all variants of differential privacy, and is written here with $(\varepsilon, \delta)$-DP for concreteness.

[4]The noise scale is $\propto \tau$ but DP methods require an additional factor of $\log(|V|)$ (due to a union bound over the support $V$) or $\log(1/\delta)$ (with approximate DP). This applies also with independent ensembles.

## 5 EMPIRICAL DEMONSTRATION

We demonstrate the benefits of coordinated ensembles (hot PATE) compared with the baseline of independent ensembles (cold PATE). For clarity and simplicity, we designed our demo so that it generates a single token. Sequential text generation performs multiple such steps. We use the Meta Llama 3 8B parameter open source language model (lla, 2024; AI@Meta, 2024).

**Generating Prompts:** We generated for each experiment $n = 10^4$ text prompts (teachers) of the following form

> On planet Z, some numbers are edible. <name> from planet
> Z eats the following numbers for breakfast: <random permu-
> tation of $C \cup \{$<private number>$\}$ > Give me an example breakfast
> number in planet Z. Respond with just the number.

The set $C$ is a fixed subset of size $|C| = k$ of the set $\mathbb{N}_{100}^{999} = \{100, \ldots, 999\}$ of the 900 3-digit numbers. We selected the set $C$ uniformly at random. The strings <name> and <private number> $\sim U[\mathbb{N}_{100}^{999} \setminus C]$ were generated separately for each prompt $i \in [n]$. For our purposes, the set $C$ is the information we want transferred whereas the prompt-specific <name> and <private number> and the ordering of $C$ in the prompt are considered sensitive. Each prompt is designed to have $k + 1$ correct answers. We report results with $k \in \{20, 100\}$. For each prompt $i \in [n]$ we retrieved the probability distribution $\boldsymbol{p}^{(i)}$ over tokens $V$ of the next-token response. Llama 3 uses a vocabulary $V$ of 128k tokens and 3-digit numbers are encoded as single tokens. The generation took a few minutes on a single A100 GPU. The distributions the model generated exhibited biases towards certain numbers and high variation. The probability of returning a 3-digit number was $0.995$ but the model generalized and returned with $25\%$ probability numbers outside the input set. Note that our aim is to transfer what the model does (including the biases and generalizing), also when it differs from the original intent of the prompt author. See Section C.1 for further details.

**Sampling vote histograms** We sampled $r = 10^3$ vote histograms $(\mathbf{c}^h)_{h=1}^r$ from each of coordinated and independent ensembles. Each histogram has total count of $n = 10^4$, since each teacher contributes one token. We use the notation $c_j^h$ for the frequency (count) of token $j$ in the $h$th histogram ($h \in [r]$).

Figures 8 and 9 visualize the average probability $\frac{1}{n} \sum_{i \in [n]} p_j^{(i)}$ of each token $j \in \mathbb{N}_{100}^{999}$ across teacher distributions. The figure also shows the average frequency $\frac{1}{r} \sum_{h=1}^r c_j^h$ over the $r = 10^3$ samples from each of independent and coordinated ensembles. This demonstrates the property (see Claim 1) that the expected number of votes for each token is the same for the two ensemble types and corresponds to the average distribution. The qualitative difference between coordinated and independent ensembles (see Claim 2) is visualized in Figure 10 by zooming on individual sampled histograms. The figure shows one sampled histogram with independent sampling and two sampled histograms with coordinated sampling. With independent sampling, frequency counts of each token $j$ are concentrated close to the expectation $\sum_i p_j^{(i)}$ and are similar across different samples and to the averages shown in Figures 8 and 9. With coordinated samples there is high variability between samples and it is possible for the frequency of a token to far exceed $\sum_i p_j^i$.

**Utility Evaluation** A token $j$ in sample $h$ can be reported in a privacy-preserving way only when its frequency exceeds the scale of the privacy noise $c_j^h > T$. We evaluate utility of coordinated and independent ensemble types by considering (i) coverage for threshold $T$: fraction of the votes that appear with token frequency at least $T$ and (ii) diversity for coverage: The number of distinct tokens that are appear with high frequency.

Figure 5 (left) shows $\mathsf{E}_h[|\{j \in V \mid c_j^h \geq T\}|]$, the average number of tokens per sample (histogram) that have frequency above $T$, for varying $T$. Observe that with independent samples, the maximum frequency $\max_{h,j \in V} c_j^h$ (over histograms and tokens) corresponds to the maximum token average probability: for $k = 20$ it is $0.14n$ and for $k = 100$ it is $0.03n$. With coordinated ensembles, the majority of samples contained a token with frequency above $0.25n$ (that is much higher than the maximum token average probability). Figure 5 (middle) reports the fraction of the votes (over samples and tokens) that are in frequencies that exceed $T$, for varying $T$. We observe that coordinated ensembles cover many more votes for a given $T$ than independent ensembles. Additionally, we

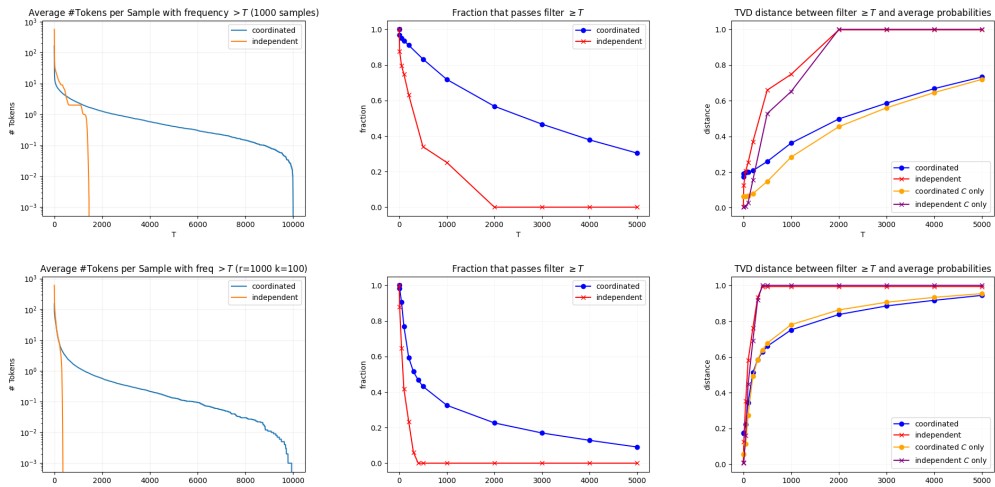

Figure 5: Left: Average number of tokens per sample with frequency filter $\geq T$. Middle: fraction of votes in frequency $\geq T$. Right: Total Variation Distance between filtered and average distribution with filter $\geq T$. Top $k = 20$ bottom $k = 100$.

observe that the coverage corresponds to the $T/n$-robust part of the distribution shown in Figure 7, that is, it corresponds to what we can hope to transfer (see Theorem 1 and Section C.2). For $k = 100$, 20% of the votes are covered with $T = 2000$ with coordinated sampling but require $T \leq 250$ with independent sampling (factor $\times 8$). For $k = 20$, 40% of votes are covered with $T = 4000$ with coordinated sampling but this coverage requires $T \leq 1000$ with independent sampling (factor $\times 4$). Independent samples have 0% coverage with $T \geq 1500$ for $k = 20$ and with $T \geq 400$ for $k = 100$. To summarize, we observe that independent ensembles have 0% coverage when $T$ exceeds the maximum average frequency whereas coordinated ensembles are effective with high $T$.

We next consider diversity per coverage. Figure 5 (right) reports the total variation distance from the average distribution. Figure 11 is a parametric plot by $T$ (not shown) that shows the relation of coverage (average number of of teacher votes over samples that occurred in counts $\geq T$) vs sparsity (number of distinct tokens that in at least one sample had count $\geq T$) with coordinated and independent ensembles. We can see that coordinated ensembles are more diverse than independent ensembles for the same coverage of votes, with an order of magnitude gap.

Figures 12 and 13 visualize the histograms of the covered votes (averaged over the $r$ samples) per token, for varying thresholds $T$. For each displayed histogram we list coverage and sparsity. Recall that the threshold $T$ corresponds to the noise scale $\sigma$ that allows for the transfer. Coverage is indicative of yield distribution and sparsity reflects lower diversity of the yield. The visualization demonstrates again the benefits of coordinated ensembles: Independent ensembles become ineffective with very low $T$, quickly losing coverage and diversity compared with coordinated ensembles. The maximum average frequency of a token was $0.14$ with $k = 20$ and $0.04$ with $k = 100$ and indeed independent ensembles transfer nothing beyond these proportions of teachers. Moreover, no generalization (shown in blue) is transferred. Coordinated ensembles on the other hand are effective also when $T$ is a fraction of teachers (20%+) that is much higher than the maximum average frequency of a token.

**Conclusion** We proposed hot PATE that enhances the PATE framework in diverse settings. Hot PATE only requires API access to proprietary models and can boost performance as a plug-in replacement to cold PATE. An important use case is in-context learning via prompts, such as generating privacy-preserving synthetic data records from sensitive records. We formally define a robust diversity-preserving aggregate of distributions and design an aggregation method that satisfies it with no privacy penalty for higher diversity. Beyond private learning, our design, with lower values of the tuneable robustness parameter, is suitable for applications such as data distillation that require robustness to few outliers or a lightweight protection against memorization but not necessarily strong privacy guarantees.

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

## A    RELATED WORK

We place our contribution in the context of prior and independent concurrent works on PATE adaptations for text generation. These works either (i) did not consider diversity or (ii) recognized it and the importance of transferring it but proposed aggregation schemes where utility decreases with diversity together with methods to limit diversity as to mitigate this perceived privacy-diversity trade-off. Our technique of ensemble coordination is an independent contribution that can enhance or replace components in some of these designs.

Tian et al. (2022) proposed a PATE extension for sequential text generation tasks in diverse settings. Their approach limited diversity: Average the teachers distributions and then truncate the tail by keeping only the top-$k$ frequencies. The work of Tang et al. (2024) (independent concurrent) took a similar approach. The distribution of each teacher is reduced to a uniform distribution over its top-$k$ token probabilities. An independent ensemble is then applied to this set of reduced distributions. This approach limits diversity to $k$ and suffers from loss of diversity while still incurring a utility trade-off with $k$. In our work, we demonstrate that averaging teacher distributions (independent ensemble) is inferior to coordinating the ensemble.

Recent prior work explored adaptations of PATE for in-context learning via prompting.

Duan et al. (2023) proposed to use each part $D_i$ of the data to create a text prompt $T_i$. The ensemble is then used to label curated queries. But while some design elements were tailored to LLMs, the workflow and privacy analysis were identical to cold PATE (Papernot et al., 2018), and in particular, did not consider diverse responses.

Wu et al. (2023) (independent concurrent work) proposed approaches to private aggregation for in-context learning with diversity. They proposed to reduce the perceived diversity in sequentially-generated text outputs by different teachers by clustering together outputs that are semantically equivalent and aggregating each cluster in a semantic space. This essentially reduces the dimensionality of the output space. The aim then is to extract and transfer this common semantics in a privacy preserving way: Map responses into a common low dimensional embedding space and privately aggregate embedding vectors or identify frequent keywords in diverse teachers' responses. The limitations are that the approach only addresses same-semantics diversity and offers no solution for semantically-distinct diverse responses and are subjected to a privacy diversity trade-off. Additionally and importantly, they require hand crafted tools to map and curate responses back and forth from a semantic space. The added value of such a mapping approach, if combined with coordinated ensembles, depends on whether the reduction of diversity that is achieved is within or across teachers. The *across* variety (see Figure 3 (B)), where the knowledge of each teacher only contains one or limited variations of the same semantic, is not eliminated by ensemble coordination and thus there is added value by addressing it via other means. The *within* variety (see Figure 3 (A)) is handled effectively by ensemble coordination and can be transferred fluidly with no privacy loss and without the need for mitigation of diversity via additional engineering. We suspect that for the in-context learning use case, and for semantic similarity that can be captured by tools external to the model (such as an embedding), the diversity eliminated is anyhow encapsulated in the base model and thus present in most teacher distributions. That is, we expect the diversity to overwhelmingly be the "within" variety.

Lin et al. (2024); Xie et al. (2024) (independent concurrent work) proposed an approach called *private evolution* for generating synthetic examples from private examples. The design used heterogeneous teachers, where each is a single private example. Initially, the base model is sampled to generate a collection of candidate (full) responses. The teachers then vote on candidates by nearest neighbor to their sensitive example in an embedding space. The next iteration then consist of a weighted sample from a privacy-preserving vote histogram. The resulting candidates are then used to generate a new set of candidates by the base model that are closer to the private distribution. This is repeated for multiple iterations. The inherent drawbacks of this approach, compared with sequential text generation, are that it is not suitable for transferring specific patterns (such as extension numbers for specific departments within an org) that are common in the private data but do not exist in the pre-training data and are not memorized by the model and can not be generalized by it. Additionally, it requires a number of candidates that is exponential in the intrinsic dimensionality of the candidate space. Therefore the realm of applications is different than Hot Pate and they are not directly comparable.

Papernot et al. (2017) (Appendix B.1) discussed using additional outputs (beyond just the noisy the maximizer) in the teachers' votes histogram for distillation tasks. They concluded that it is beneficial for utility but does not justify the privacy loss. Despite the superficial resemblance, this is very different from what we do as we capture diversity in the generation of the histogram where we "force" the teachers to agree but there is a distribution on the agreement token.

Finally, there are multiple innovative adaptations of PATE to non-categorical settings (aggregate vectors rather than labels) applied with generative models. The works we are aware of address different problems and use different techniques than hot PATE. For example, image generation using generative adversarial networks (GAN): Jordon et al. (2018) proposed to train student discriminator using a cold-PATE like labeling approach. Long et al. (2021) proposed to train a student generator by aggregating the gradients produced by teachers discriminators. Notably, as with hot PATE, this design does not require external generation of examples in order to facilitate transfer. Instead, it uses the built-in property of generators to produce examples from random strings.

## B  PROPERTIES OF COORDINATED ENSEMBLES

*Proof of Claim 2.* The first statement in the claim follows from the denominator satisfying

$$1 \leq \sum_j \max\{p_j^{(i)}, p_j^{(k)}\} \leq 2 - \max\{p_j^{(i)}, p_j^{(k)}\} \leq 2 \, . \tag{3}$$

The inequality follows using the more refined upper bound (3) on the denominator. □

The overall agreement probability of the two teachers (over all tokens) is the (weighted) Jaccard index (Jaccard, 1901) of the distributions:

$$\Pr_{\boldsymbol{y} \sim Y_{\mathrm{coo}}} [y_i = y_k] = \frac{\sum_j \min\{p_j^{(i)}, p_j^{(k)}\}}{\sum_j \max\{p_j^{(i)}, p_j^{(k)}\}} \, .$$

In particular, when two teacher distributions are identical, the samples are the same

$$\boldsymbol{p}^{(i)} = \boldsymbol{p}^{(k)} \implies \Pr_{\boldsymbol{y} \sim Y_{\mathrm{coo}}} [y_i = y_k] = 1.$$

We establish the claim in Theorem 1. We show that a token $j$ for which $m$ teachers $i$ have $p_j^{(i)} > q$ has frequency at least $m/2$ with probability at least $0.34q$. This follows by substituting $p = 1/2$ in the following more general claim:[5]

**Lemma 1** (diversity transfer). *For any token $j$ and $p, q \in [0, 1]$,*

$$\Pr_{\boldsymbol{c} \sim H(Y_{\mathrm{coo}})} \left[ c_j \geq \left\lfloor p \cdot \sum_{i \in n} \mathbf{1}\{p_j^{(i)} \geq q\} \right\rfloor \right] \geq \frac{1}{2} \ln(1/p) q \, .$$

*Proof.* Let $i$ be such that $p_j^{(i)} \geq q$. Fix the sampled min value $x \sim \mathsf{Exp}[q]$ for $q$ part of the probability of $j$. The distribution of the remaining part is $y \sim \mathsf{Exp}[1 - p_j^{(i)}]$ which is stochastically smaller than $\mathsf{Exp}[1 - q]$. We get that

$$\Pr[y_i = j] \geq \Pr_{y \sim \mathsf{Exp}[1-q]} [y > x] = e^{-x(1-q)} \, .$$

Fix $p \in [0, 1)$. It follows that the probability that $\Pr[y_i = j]$, conditioned on $x < \frac{-\ln p}{1-q}$ is at least $e^{-x(1-q)} \geq p$. The respective random variables $y_i$ on different teachers that may share part of the distribution can only be nonnegatively correlated. Therefore, if there are $c_{j,q}$ teachers with $p_j^{(i)} \geq q$ then the distribution of the number of teachers with $y_i = j$ is stochastically larger than $\mathsf{Bin}[e^{-x(1-q)}, c_{j,q}]$, which for any $x \leq \frac{-\ln p}{1-q}$ is stochastically larger than $\mathsf{Bin}[p, c_{j,q}]$. The median of

---

[5]The general statement allows for different tradeoffs between $\beta$ and the threshold in Theorem 1

the Binomial distribution $\mathsf{Bin}[p, c_{j,q}]$ with probability at least $1/2$ is larger than $\lfloor pc_{j,q} \rfloor$. Therefore, with this conditioning on $x$, there are at least $\lfloor pc_{j,q} \rfloor$ teachers with $y_i = j$.

$$\Pr_{(y_i)_{i \in [n]} | x < \frac{-\ln p}{1-q}} [c_j \geq \lfloor pc_{j,q} \rfloor] \geq 1/2 \ . \tag{4}$$

The event $x < \frac{-\ln p}{1-q}$ occurs with probability at least

$$\Pr_{x \sim \mathsf{Exp}[q]} [x < \frac{-\ln p}{1-q}] = 1 - e^{(\ln p)q/(1-q)} \geq -(\ln p)q \ .$$

Combining with (4), we obtain the claim in the statement of the Lemma. $\qquad \square$

To establish relevance we show that high frequency must have a "backing." The following is immediate from (2) and Markov's inequality (and is tight in the sense that for any $T$ there are distributions where equality holds):

**Lemma 2** (relevance)**.** *For any token $j$ and $T$,*

$$\Pr_{\mathbf{c} \sim H(Y_{\mathrm{coo}})} [c_j \geq T] \leq \frac{1}{T} \sum_{i \in [n]} p_j^{(i)} \ .$$

## C  FURTHER DETAILS ON EMPIRICAL DEMONSTRATION

### C.1  PROPERTIES OF THE GENERATED DISTRIBUTIONS

The distributions deviated from a balanced response over correct answers: The model exhibited bias towards certain numbers and spurious dependencies on private components. Our evaluation is of the effectiveness of transferring the knowledge of the model *as reflected in the generated response distributions*. We observed the following:

- The probability assigned by the model to tokens that are not 3-digit numbers is negligible: The average probability (over teachers) of a response token in $\mathbb{N}_{100}^{999}$ was $\mathsf{E}_{i \in [n]} \sum_{j \in \mathbb{N}_{100}^{999}} p_j^i \approx 0.997$ for $k = 20$ and $\approx 0.994$ for $k = 100$.

- Tokens in $C$ dominate but other 3-digit numbers are likely: The average probability of a token in $C$ was $\mathsf{E}_{i \in [n]} \sum_{j \in C} p_j^i \approx 0.716$ ($k = 20$ tokens) and $\approx 0.75$ ($k = 100$). Recall that only one in $k$ numbers in the prompt was in $\mathbb{N}_{100}^{999} \setminus C$, therefore the probability of $25\%+$ assigned to these tokens is explained by the model generalizing that additional 3-digit numbers are edible on Planet Z.

- Despite symmetric prompt construction, there is significant variability in the average probability of different tokens in $C$ and in the probability across teachers of the same token. This is an artifact of the model. Figure 6 reports the average (over prompts) of the probability of each token and demonstrates variability between tokens. The error bars indicate variability in the token probability across teachers.

### C.2  QUANTIFYING HOW MUCH IS TRANSFERABLE

**Remark 3** (Robust Average)**.** *We use the $\tau$-robust part of the average of the teachers distributions as an indicative upper bound on what can be potentially privately transferred:*

$$P_j(\tau) := \frac{1}{n} \sum_{i \in [n]} \min \left\{ p_j^{(i)}, (\{p_j^{(h)}\}_{h \in [n]})_{(\tau)} \right\} \textit{ for } j \in V \tag{5}$$

*where $(\{p_j^{(h)}\}_{h \in [n]})_{(\tau)}$ is the $\tau$th largest probability of token $j$ in a teacher distribution. Note that $(P_j(1))_{j \in V}$ is the average distribution and the values are non-increasing with $\tau$. We also consider the $\tau$-robust probability mass defined as $P(\tau) := \sum_{j \in V} P_j(\tau) \leq 1$. The complement $1 - P(\tau)$ is indicative lower bound on the probability of $\perp$ in the robust aggregate.*

Figure 7 reports the $\tau$-robust fraction of the average distribution for varying $\tau$ (see Remark 3). This is the part of the average distribution that we can hope to transfer via coordinated ensembles with support $\tau$. Recall that variability in the same token among teachers decreases transferability whereas variability among tokens does not.

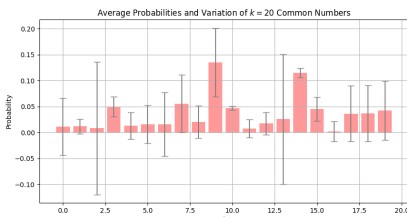 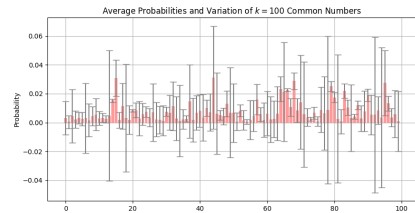

Figure 6: Average probability, over teachers, of the $k$ tokens in $C$ (left is $k = 20$, right is $k = 100$). The error bars indicate the contribution of the token to the average total variation distance over pairs of teacher distributions.

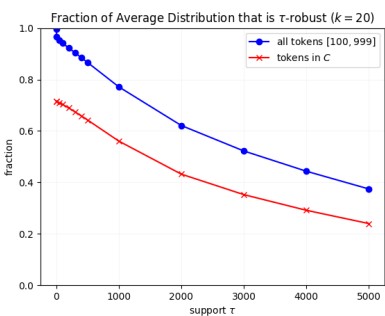 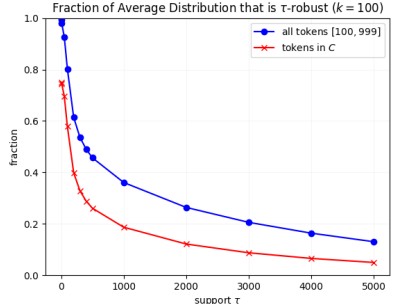

Figure 7: The $\tau$-robust part of the distribution for varying $\tau$ (see Remark 3). Left is $k = 20$ right is $k = 100$.

### C.3 INDEPENDENT VERSUS COORDINATED HISTOGRAMS

The marginal distribution is the same (Figures 8 and 9) but coordinated histograms are not concentrated around their expectation (Figure 10)

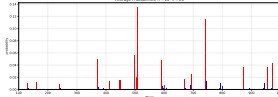 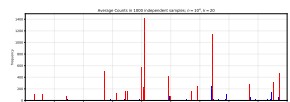 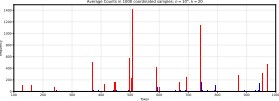

Figure 8: $k = 20$: For all tokens (tokens in $C$ shown in read): Average probability over teachers (left). Average frequency of $r = 1000$ samples using independent (middle) and coordinated (right) ensembles.

## D AGGREGATION METHODS OF FREQUENCY HISTOGRAMS

Our aggregation methods are applied to frequency histograms generated by a coordinated ensemble and return a token or $\perp$. We propose two meta schemes that preserves diversity in the sense of Definition 1: One for homogeneous ensembles, where we use $\tau > n/2$, in Section D.1 and one for heterogeneous ensembles, where $\tau \ll n/2$ (but large enough to allow for DP aggregation), in Section D.2. To establish diversity preservation, we consider the end-to-end process from the teacher distributions to the aggregate distribution. To establish privacy, it suffices to consider the histogram in isolation, as it has the same sensitivity as vote histograms with cold PATE: When one teacher distribution changes, one token can gain a vote and one token can lose a vote. Noting that the shared randomness $\rho$ is considered "public" data. We then explore (Sections E and F) DP implementations that admit data-dependent privacy analysis so effectively many more queries can be performed for the same privacy budget. We can avoid privacy loss on responses that agree with the prior distribution of the public model with a public prompt. We can benefit from the particular structure of histograms generated by coordinated ensembles. The privacy loss does not depend on queries with no yield, with

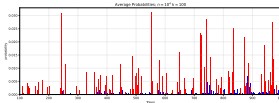 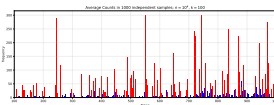 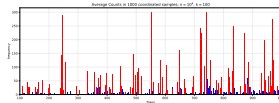

Figure 9: $k = 100$: For all tokens (tokens in $C$ shown in read): For all tokens (tokens in $C$ shown in read): Average probability over teachers (left). Average frequency of $r = 1000$ samples using independent (middle) and coordinated (right) ensembles.

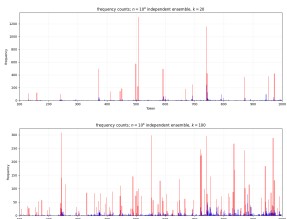 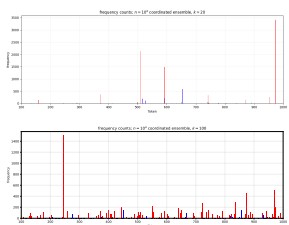 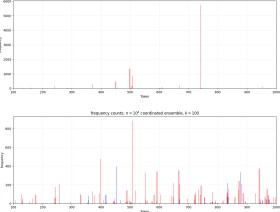

Figure 10: Frequency counts per token in individual sampled histograms. Left: Independent ensemble. Middle and Right: Coordinated ensemble. Top $k = 20$ bottom $k = 100$.

high agreement, or with agreement with a public prior. With heterogeneous ensembles we can also gain from individualized per-teacher privacy charging.

### D.1 HOMOGENEOUS ENSEMBLES

---

**Algorithm 2:** `DistAgg` homogeneous

$\boldsymbol{c}, \rho \leftarrow \texttt{CoordinatedSamples}((\boldsymbol{p}^{(i)})_{i \in [n]})$          // Algorithm 1

$(j, \hat{c}_j) \leftarrow \texttt{NoisyArgMax}_L(\boldsymbol{c})$          // DP noisy maximizer with error $L$

**if** $\hat{c}_j > (n/2 + L)$ **then return** $j$ **else return** $\perp$

---

When $\tau > n/2$, there can be at most one token $j$ with frequency $c_j \geq \tau$. If there is such a token, we aim to report it. Otherwise, we return $\perp$. Our scheme is described in Algorithm 2 in terms of a noisy maximizer (`NoisyArgMax`$_L$) procedure. The latter is a well studied construct in differential privacy (McSherry and Talwar, 2007; Durfee and Rogers, 2019; Qiao et al., 2021). Generally, methods vary with the choice of noise distribution and there is a (high probability) additive error bound $L$ that depends on the privacy parameters and in some cases also on the support size and confidence. For our purposes, we abstract this as `NoisyArgMax`$_L$ that is applied to a frequency histogram $\boldsymbol{c}$ and returns $(j, \hat{c}_j)$ such that $|c_j - \hat{c}_j| < L$ and $\max_{h \in V} c_h - c_j \leq 2L$. We show that the method is diversity preserving:

**Lemma 3** (Diversity-preservation of Algorithm 2). *For $\mu > 1$, Algorithm 2, instantiated with* `NoisyArgMax`$_L$ *as described, is diversity preserving in the sense of Definition 1 with $\tau = \mu(n/2 + 2L)$, $\beta = \ln(\mu)/2$ and $\gamma = 2$.*

*Proof.* We apply Lemma 1 with $p = 1/\mu$. We obtain that the token $j$ has frequency at least $c_j \geq n/2 + 2L$ with probability at least $0.5 \ln(\mu)q$. Therefore we have $\hat{c}_j \geq n/2 + L$ with probability at least $0.5 \ln(\mu)q$. Note that a token can only be reported if its frequency is $c_j > n/2$. Using $T = n/2$ in Lemma 2 we obtain that the relevance requirement is satisfied with $\gamma = 2$. $\qquad\square$

The two most common noise distributions for DP are Gaussian and Laplace noise. (Cold) PATE was studied with both. The Gaussian-noise based Confident-GNMax aggregator (Papernot et al., 2018; Duan et al., 2023) empirically outperformed the Laplace-based LNMAX (Papernot et al., 2017) on cold PATE. The advantages of Gaussian noise are concentration (less noise to separate a maximizer from low frequency tokens), efficient composition, and more effective data dependent privacy analysis. Laplace-based noise on the other hand can preserve sparsity (a consideration as the

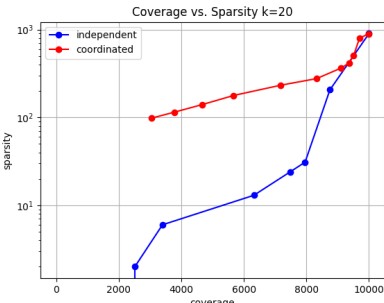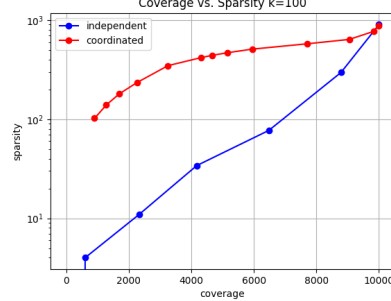

Figure 11: Coverage (average across samples of the number of $n$ teacher votes that passed count filter $T$) versus sparsity (number of distinct tokens that at least in one sample had count $\geq T$) with coordinated and independent ensembles, when sweeping the parameter $T$ (not shown). $k = 20$ (left) and $k = 100$ (right).

$T = 100$

coverage: coo: 94% ind: 75%

sparsity: coo: 417 ind: 24

$T = 200$

coverage: coo: 91% ind: 63%

sparsity: coo: 365 ind: 13

$T = 500$

coverage: coo: 83% ind: 34%

sparsity: coo: 277 ind: 6

$T = 1000$

coverage: coo: 72% ind: 25%

sparsity: coo: 233 ind: 2

$T = 2000$

coverage: coo: 57% ind: 0%

sparsity: coo: 178 ind: 0

$T = 5000$

coverage: coo: 30% ind: 0%

sparsity: coo: 98 ind: 0

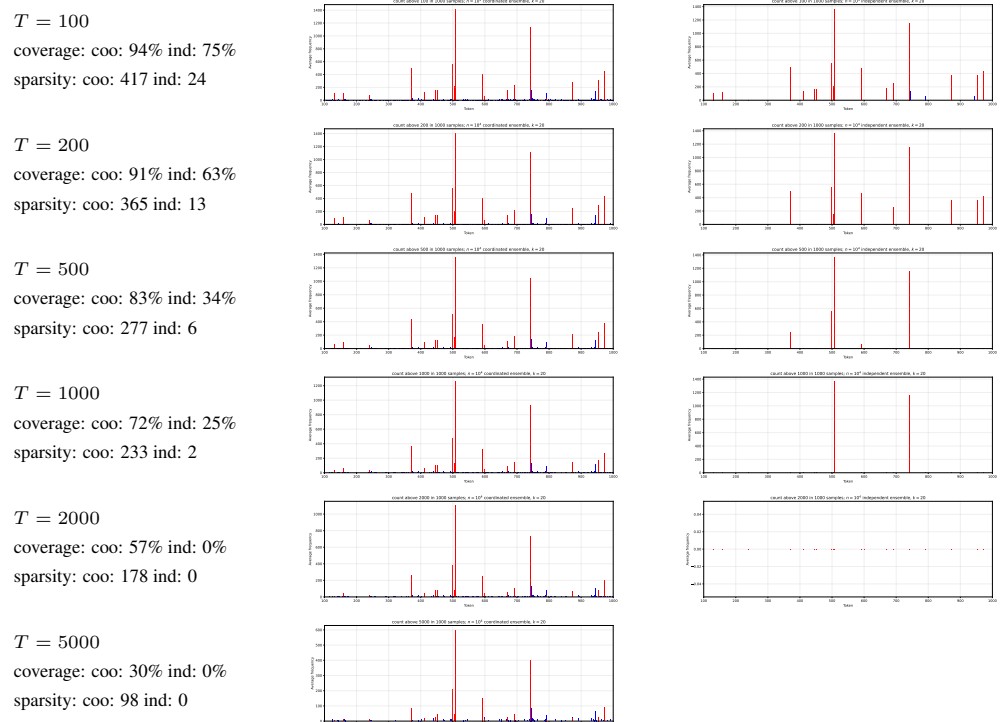

Figure 12: Coverage histograms averaged over $r = 10^3$ samples. Filter $T \in [100, 200, 500, 1000, 2000, 5000]$. $k = 20$. Left: Coordinated. Right: Independent.

key space of tokens or strings of token can be quite large), there is an optimized mechanism with weighted sampling, and there are recent improvement on data-dependent privacy analysis across many queries (the situation with hot PATE) (Cohen and Lyu, 2023). Our privacy analysis in Section E uses a data-dependent Laplace-based approach.

### D.2 HETEROGENEOUS ENSEMBLES

For lower values of $\tau$, we propose the meta-scheme described in Algorithm 3: We perform weighted sampling of a token from $c$ and return it if its count exceeds $2L$. If it is below $2L$ we may return either $j$ or $\perp$. We propose DP implementations in Section F. We establish that Algorithm 3 is diversity-preserving:

$T = 50$

coverage: coo: 90% ind: 65%

sparsity: coo: 638 ind: 77

$T = 100$

coverage: coo: 77% ind: 42%

sparsity: coo: 576 ind: 34

$T = 200$

coverage: coo: 59% ind: 23%

sparsity: coo: 509 ind: 11

$T = 300$

coverage: coo: 52% ind: 6%

sparsity: coo: 468 ind: 4

$T = 400$

coverage: coo: 47% ind: 0%

sparsity: coo: 441 ind: 0

$T = 1000$

coverage: coo: 33% ind: 0%

sparsity: coo: 347 ind: 0

$T = 2000$

coverage: coo: 23% ind: 0%

sparsity: coo: 234 ind: 0

$T = 5000$

coverage: coo: 9% ind: 0%

sparsity: coo: 102 ind: 0

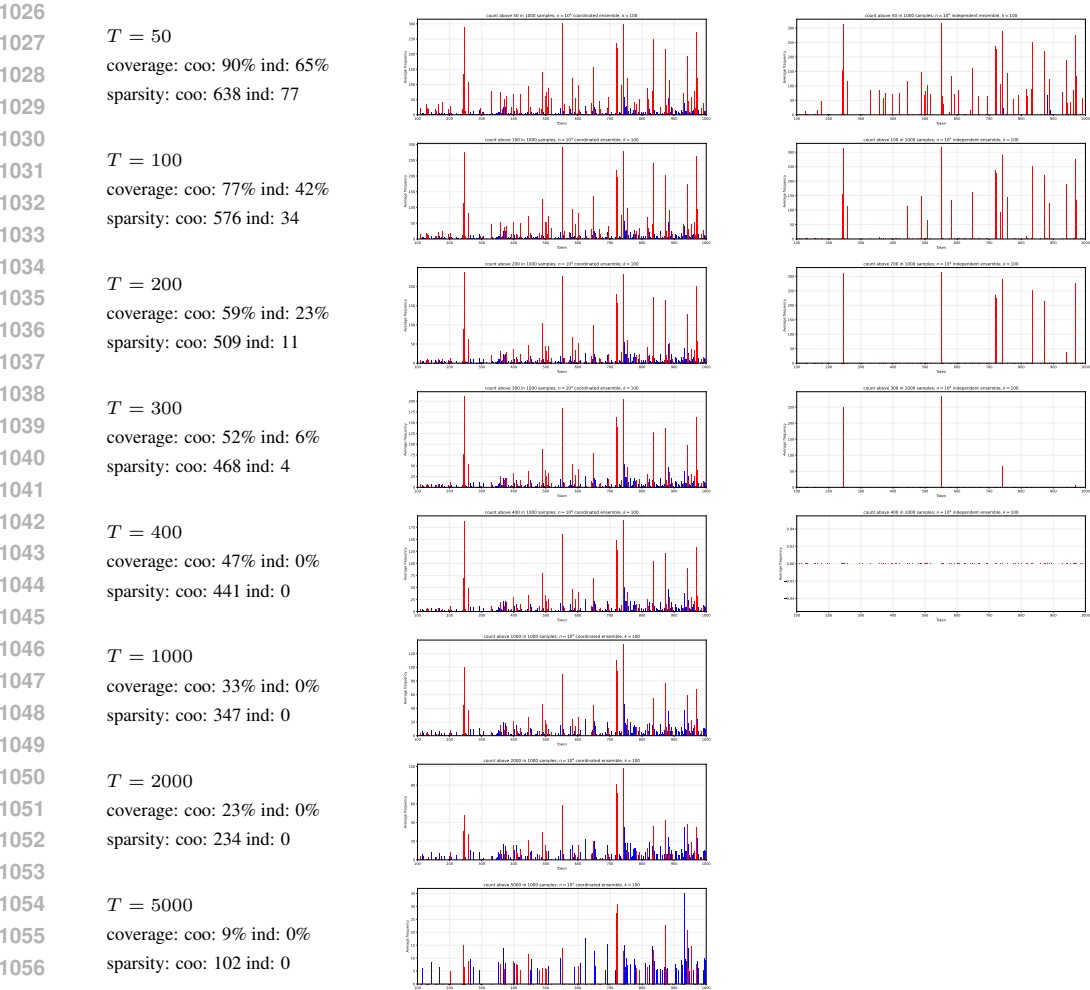

Figure 13: Coverage histograms averaged over $r = 10^3$ samples. Filter $T \in [50, 100, 200, 300, 400, 1000, 2000, 5000]$. $k = 100$ Left: coordinated Right: Independent

---

**Algorithm 3:** `DistAgg` Heterogeneous

---

$\boldsymbol{c}, \rho \leftarrow \texttt{CoordinatedSamples}((\boldsymbol{p}^{(i)})_{i \in [n]})$        `// Algorithm 1`

Sample $j \in V$ with probability $\frac{c_j}{n}$        `// Weighted sampling of a token from c`

**if** $c_j \geq 2L$ **then return** $j$ **else return** $j$ *or* $\perp$

---

**Lemma 4** (Diversity-preservation of Algorithm 3). *For $\mu > 1$, Algorithm 3 is diversity preserving in the sense of Definition 1 with $\tau = \mu 2L$, $\beta = \frac{1}{2\mu} \ln(\mu)$ and $\gamma = 1$.*

*Proof.* Consider the first requirement of Definition 1. Consider a token $j$ with $c_{j,q} \geq \tau$. From Lemma 1 using $p = 1/\mu$ we obtain that the token $j$ has frequency at least $c_j \geq c_{j,q}/\mu \geq 2L$ with probability at least $0.5 \ln(\mu)q$. The token is sampled with probability $\min\{1, kc_j/n\}$ and if so appears also in $\boldsymbol{c}^*$ (since $c_j \geq 2L$). The expected size (number of entries) of $\boldsymbol{c}^*$ is at most $k$ and thus it is returned if sampled with probability at least $1/k$. Overall it is sampled and reported with probability at least $\min\{1/k, c_j/n\}$. In total, the probability is $P_j \geq \min\{1/k, c_{j,q}/(\mu n)\}0.5 \ln(\mu)q \geq \frac{1}{2k\mu} \ln(\mu) \frac{c_{j,q}}{n} q$.

The second requirement of Definition 1 is immediate. The expected frequency of token $j$ is $\sum_{i \in [n]} p_j^{(i)}$ and it is sampled with probability at most $\frac{k}{n} \sum_{i \in [n]} p_j^{(i)}$. It can only be the output if sampled.    $\square$

# E   PRIVACY ANALYSIS CONSIDERATIONS

The effectiveness of Hot PATE depends on the number of queries with yield (token returned) that can be returned for a given privacy budget. In this section we explore the benefits of data-dependent privacy analysis when the aggregation follows Algorithm 2 (homogeneous ensembles). We use synthetically generated teacher distributions with varying size common component (that can be arbitrarily diverse) and distinct (private) components.

Broadly speaking, with data-dependent analysis, we incur privacy loss on "borderline" queries where the output of the DP aggregation has two or more likely outputs. Queries that return a particular token with high probability or return $\perp$ with high probability incur little privacy loss.

We demonstrate that with Algorithm 2, we can expect that only a small fraction of frequency histograms generated by coordinated ensembles are "borderline." (i) For queries with high *yield* (high probability of returning a token over the sampling of the shared randomness), the generated histograms tend to have a dominant token (and thus lower privacy loss). This because coordinated ensembles tend to "break ties" between tokens. (ii) For queries with low yield (high probability of $\perp$ response and low probability of returning a token), the total privacy loss only depends on yield responses. This means that high $\perp$ probability does not cause performance to deteriorate.

This is important because both these regimes are likely in sequential text generation and with coordinated ensembles. We expect many of the tokens to follow the base model distribution and therefore have high agreement and not incur privacy loss. Or alternatively, instructions that require private data have no agreement and return $\perp$. The dependent privacy analysis means that generally we can process many more queries for the privacy budget than if we had just used a DP composition bound.

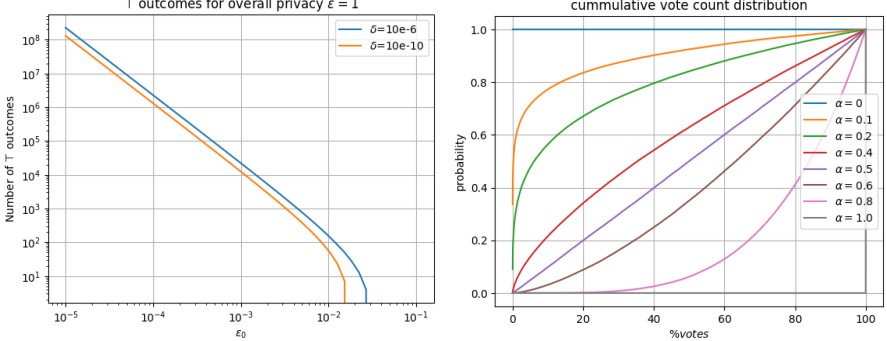

Figure 14: Left: Number of $\top$ responses for $\varepsilon_0$-DP queries for total $\varepsilon = 1$ loss. Right: Cummulative maximum frequency for varying common part $\alpha$.

Our evaluation here uses $(\varepsilon, \delta)$ differential privacy (Dwork et al., 2006):

**Definition 2** (($\varepsilon, \delta$)-Differential Privacy). A randomized mechanism $\mathcal{M}$ provides $(\varepsilon, \delta)$-differential privacy if, for any two datasets $D$ and $D'$ differing in at most one element, and for any subset of outputs $S \subseteq \text{Range}(\mathcal{M})$, the following holds:

$$\Pr[\mathcal{M}(D) \in S] \leq e^\varepsilon \Pr[\mathcal{M}(D') \in S] + \delta.$$

Concretely we consider `NoisyArgMax` using (Cohen et al., 2021) [6] with the maximum sanitized frequency, with privacy parameters $(\varepsilon_0, \delta_0)$. For privacy analysis across queries we applied the Target Charging Technique (TCT) of Cohen and Lyu (2023) with the *boundary-wrapper* method. The wrapper modifies slightly the output distribution of the query algorithm (after conditioning on $\rho$!) to include an additional outcome $\top$ (*target*). The wrapper returns $\top$ with this probability (that depends on the response distribution) and otherwise returns a sample from the output distribution of the wrapped algorithm. The probability of $\top$ is at most $1/3$ and decreases with agreement (vanishes

---

[6]We mention the related (non optimized) sparsity-preserving methods (Bun et al., 2019; Korolova et al., 2009; Vadhan, 2017) and optimized but not sparsity-preserving (Ghosh et al., 2012).

when there is response with probability closer to 1). The technique allows us to analyse the privacy loss by only counting target hits, that is, queries with $\top$ response. Since the probability of $\top$ is at most $1/3$, we get in expectation at least two useful responses per target hit. But in case of agreements, we can get many more. Figure 14 (left) reports the number of $\top$ (target) responses we can have with the boundary wrapper method as a function of $\varepsilon_0$ with overall privacy budget is $\varepsilon = 1$. When $\varepsilon_0 \leq 0.01$, it is about $(10\varepsilon_0)^{-2}$.

With hot PATE, we are interested in *yield* responses, those that return a token (not $\bot$, and when we apply the boundary wrapper, also not $\top$). We study how the yield probability behaves for histograms generated by coordinated ensembles.

**Synthetic Teacher distributions:** We parametrize the set of teacher distributions by $\alpha \in (0, 1]$, which is the probability of a common part to all distribution. This component is what we aim to transfer to the student. The teacher distributions have probability vectors of the form

$$\boldsymbol{p}^{(i)} = \alpha \cdot \boldsymbol{s} + (1 - \alpha) \cdot \boldsymbol{r}^{(i)} \ ,$$

where $\boldsymbol{s}$ and $\boldsymbol{r}^{(i)}$ are probability vectors. That is, with probability $\alpha$ there is a sample from the common distribution $\boldsymbol{s}$, and with probability $(1 - \alpha)$, there is a sample from an arbitrary distribution that is specific to each teacher. Note that the common component $\boldsymbol{s}$ can be arbitrarily diverse, that is, $\|\boldsymbol{s}\|_1$ is permitted to be arbitrarily small.

When the histogram is generated by a coordinated ensemble, then the distribution of the maximum frequency $c$ of a token is dominated by sampling $y \sim \mathsf{Exp}[\alpha]$ and then $c \sim \mathsf{Bin}[e^{-y \cdot (1-\alpha)}, n]$. It is visualized in Figure 14 (right) for varying values of $\alpha$. Note that across all weights $\alpha > 0$ of the shared component, no matter how small $\alpha$ is, there is probability $\approx \alpha$ of being above a high threshold (and returning a token). The probability of $\bot$ (no agreement) in this case can be $\approx 1 - \alpha$. Therefore $\alpha$ parametrizes the probability of yield over the sampling of the shared randomness.

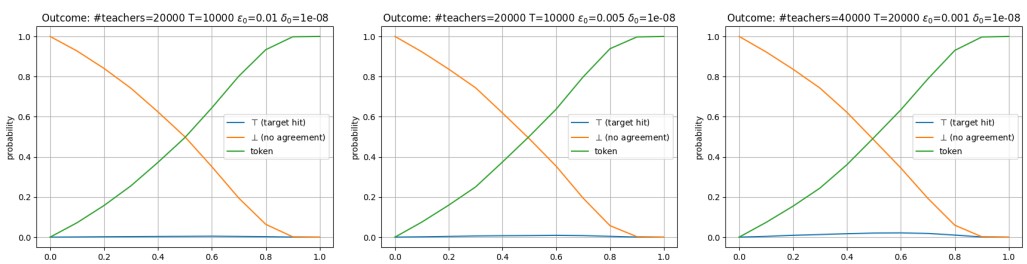

Figure 15: Sweep of $\alpha$, showing probabilities of outcomes: token, $\bot$, $\top$ (target hit).

Figure 15 shows the distribution of responses as we sweep $\alpha$, broken down by $\top$ (target hit), $\bot$ (abort), and token (yield). The number of queries we process per target hit, which is the inverse of the probability of $\top$, is $\gtrsim \varepsilon_0 n$. It is lowest at $\alpha \approx T/n$ and is very high for small and large $\alpha$, meaning that the privacy cost per query is very small.

The yield (probability of returning a token) per query is $\approx \alpha$. Note that as $\alpha$ decreases, both yield and target probabilities decrease but their ratio remains the same: In the regime $\alpha \leq T/n$, the yield per target hit is $\approx \varepsilon_0 n/2$. Queries with $\alpha \gg T/n$ are essentially free in that the yield (token) probability is very high and the $\top$ (target hit) probability is very low.

When using $n = C_\delta / \varepsilon_0$ ($C_\delta \approx 2 \log(1/\delta_0)$ teachers and plugging this in, we obtain that we get $\gtrsim 0.005 \frac{1}{C_\delta} n^2$ yields for overall privacy budget $\varepsilon = 1$. This means that we pay only for yield and not for queries. Note that this holds in the "worst case" across all $\alpha$ values, but the number of yields can be much higher when queries have large $\alpha$ (and "yields" do not incur privacy loss).

## F DP METHODS FOR HETEROGENEOUS ENSEMBLES

We propose two DP methods to implement Algorithm 3 (Section D.2) with different trade offs. In both cases we can apply data-dependent privacy analysis so that queries that do not yield a token

(that is, return $\perp$) are essentially "free" in terms of the privacy loss. The parameter $L$ depends on the privacy parameters (and logarithmically on $|V|$).

Importantly, with the second method we can apply privacy analysis with individual charging, where instead of charging the whole ensemble as a unit we only charge teachers that contributed to a response. With heterogeneous ensembles we expect the diversity to arise both from individual distributions and from differences between teachers and therefore with individual charging allows for much more efficient privacy analysis when different groups of teachers support each prediction.

**Private Weighted Sampling**    This method gains from sparsity (histogram support being much smaller than $|V|$) but the calculation of privacy loss is for the whole ensemble. We can do the analysis in the TCT framework (Cohen and Lyu, 2023) so that privacy loss only depends on yield queries (those that return a token). We perform weighted sampling by frequency of each token to obtain the sampled histogram $\boldsymbol{c}'$ and then sanitize the frequencies of sampled tokens using the end-to-end sparsity-preserving method of Cohen et al. (2021) to obtain $\boldsymbol{c}^*$. The sanitizing prunes out some tokens from $\boldsymbol{c}'$ with probability that depends on the frequency $c_j$, privacy parameters, and sampling rate. All tokens in $\boldsymbol{c}'$ with frequency above $2L$, where $L$ only depends on the privacy parameters, remain in $\boldsymbol{c}^*$.[7] The final step is to return a token from $\boldsymbol{c}^*$ selected uniformly at random or to return $\perp$ if $\boldsymbol{c}^*$ is empty.

**Individual Privacy Charging**    This method does not exploit sparsity, but benefits from individual privacy charging (Kaplan et al., 2021; Cohen and Lyu, 2023). It is appropriate when $2L \ll n$. The queries are formulated as counting queries over the set of teachers. The algorithm maintain a per-teacher count of the number of counting queries it "impacted." A teacher is removed from the ensemble when this limit is reached. Our queries are formed such that at most $O(2L)$ teachers (instead of the whole ensemble) can get "charged" for each query that yields a token.

To express Algorithm 3 via counting queries we do as follows: We sample a sampling rate $\nu \sim U[1/n, 1]$ of teachers and sample a token $v \in V$ uniformly. We sample the teachers so that each one is included with probability $\nu$ and count the number $c'_v$ of sampled teachers with $y_i = v$. We then do a `BetweenThresholds` test on $c'_j$ (using (Cohen and Lyu, 2023) which improves over Bun et al. (2017)) to check if $c'_v \geq 2L$. For "above" or "between" outcomes we report $v$. If it is a "between" outcome we increment the loss counter of all sampled teachers with $y_i = v$ (about $2L$ of them). We note that this process can be implemented efficiently and does not require explicitly performing this "blind" search.

Teachers that reach their charge limit get removed from the ensemble. The uniform sampling of the sampling rate and token emulates weighted sampling, where the probability that a token gets selected is proportional to its frequency. The sub-sampling of teachers ensures that we only charge the sampled teachers. Teachers are charged only when the query is at the "between" regime so (with high probability) at most $\approx 2L$ teachers are charged. Because we don't benefit from sparsity, there is overhead factor of $\log(|V|(n/L))$ in the privacy parameter (to bound the error of this number of queries) but we gain a factor of $n/L$ by not charging the full ensemble for each query in the heterogeneous case where most teachers have different "solutions" to contribute.

---

[7]We note that the method also produces sanitized (noised) frequency values $c^*_j$ for tokens in $\boldsymbol{c}^*$ such that $|c^*_j - c_j| \leq L$. And hence can also be used for `NoisyArgMax`

