# OpenReview forum: "Hot PATE: Private Aggregation of Distributions  for Diverse Tasks"
_ICLR.cc/2025/Conference — Submitted to ICLR 2025_

### Official Review · Reviewer_QVdG · 2024-11-01

**Soundness:** 3
**Presentation:** 2
**Contribution:** 2
**Rating:** 6
**Confidence:** 4

**Summary:**

This paper introduces "hot" PATE, an extension of PATE designed for in-context learning via prompts, addressing tasks that are "diverse" and open-ended. They empirically demonstrate the potential of hot PATE for in-context learning.

**Strengths:**

1. The motivation is clear: sequential text generation tasks through in-context learning are inherently diverse ("hot") with multiple valid responses.


2. The idea of aggregating responses from different teachers to maintain both diversity and privacy is interesting.

**Weaknesses:**

1. My primary concern is the empirical evaluation. The utility of in-context learning is typically measured by accuracy in the literature (e.g., [1,2,3]). However, this paper does not report in-context learning accuracy on specific tasks. It is unclear how much benefit hot PATE can provide for in-context learning. Additionally, the experiment is conducted on only one dataset, which is insufficient, and there is only one baseline ("cold" PATE). It is unclear why comparisons to prior in-context learning work (e.g., [1,2,3]) are not included.


[1] Duan, Haonan, et al. "Flocks of stochastic parrots: Differentially private prompt learning for large language models." Advances in Neural Information Processing Systems 36 (2024).

[2] Tang, Xinyu, et al. "Privacy-Preserving In-Context Learning with Differentially Private Few-Shot Generation." The Twelfth International Conference on Learning Representations.

[3] Wu, Tong, et al. "Privacy-Preserving In-Context Learning for Large Language Models." The Twelfth International Conference on Learning Representations.


2. The paper states that Wu et al. (2023), Lin et al. (2024), and Xie et al. (2024) are independent concurrent work, which is inaccurate. These should be considered prior work, as Wu et al. (2023) and Lin et al. (2024) were published at ICLR 2024, and Xie et al. (2024) at ICML 2024.


3. I suggest extending the literature review of this paper by including the work "Tang, Xinyu, et al. Privacy-Preserving In-Context Learning with Differentially Private Few-Shot Generation. The Twelfth International Conference on Learning Representations.". This work studies differentially private in-context learning and proposes to use the sample and aggregate framework to generate DP synthetic examples for in-context learning inference. It could also serve as an experimental baseline for comparison.




3. Some typos:

(1) I recommend ensuring the correct application of \citet and \citep.


(2) Missing periods in Line 299, Line 396, and Line 427.

**Questions:**

As in the weaknesses.

---

> ### Author Response · Authors · 2024-11-16
> **Response to the review**
>
> Thank you for your review! We plan to upload a revised version soon. Here is our response to questions:
>
> ## Question 1.
>
> You are referring to accuracy on very specific benchmarks or setups. We believe that such an empirical evaluation is not relevant or needed here. Let us explain:
>
> The application we consider is sequential text generation via PATE, and our method, of coordinated ensembles, offers a very significant improvement.  This is established mathematically, and an order of magnitude improvement is demonstrated empirically over the baseline of independent ensembles. For a given privacy budget, the distribution of the hot PATE generated text is much closer to the original teachers distributions than that of cold PATE generated text. This is the ultimate metric for the problem we are out to solve. It is not about a specific task or set of tasks. The improvement factors in whenever the response is diverse (higher entropy, multiple good answers). The baseline of independent ensembles is the method used in [1] and some other works including [2] that you mention.
>
> The relation of the related works [1],  [3] is explained in Appendix A. These works used PATE (sample an aggregate) in a different way.  The relevance of [3] is that they acknowledged diversity and the perceived tradeoff with utility and used clustering in the semantic space to reduce it. Our approach shows that it may not be necessary to do so as inherently there is no tradeoff. In any case, it is not a pure PATE sequential text generation approach.
>
> As for [2] (concurrent work) – the version we will upload adds a citation to [2]. Thank you! It appears that the method of [2] is to limit diversity by trimming to top-k tokens from each teacher and taking a uniform distribution over these tokens. This appears to be an attempt to limit diversity in order to counter the cold PATE tradeoff we demonstrate. But this is inferior to coordinating the ensemble as it fixes k and does not distinguish among the top-k. It also modifies the distributions and makes them further from that of the tuned LLM, which we believe is less desirable. Moreover, observe that our method of coordination would improve utility vs privacy even with the modified teacher distributions used in [2]. Again, it is provably better with coordination, and that component in their work can be replaced by our method of coordinating the ensemble. There is no need for experiments to validate this because it is provably more effective: Their approach satisfied definition 1 only in very limited settings.
>
> Finally, again, we did not build a system that competes with other systems. Nor do we claim to. We propose a method that is mathematically established to be used with all such systems and include a demo to showcase its benefits. It can be plugged in with any existing or future system that applies PATE for sequential text generation. The existence of all these related/prior/concurrent/future works on using PATE with prompts only shows that our work is relevant.
>
> ## Question 2
>
> A version of our work (that presented the mathematical framework with motivation, but no Llama experiments) was made public around the same time or before. Therefore, these works are concurrent and not prior. Regardless, none of these other works proposed coordinated ensembles and in some cases, the works could benefit from it. So they do not subsume our work.
>
> ## Question 3
>
> We will include a review of [2] in the revised version and thank you for pointing it out. But as explained, the experimental evaluation on the particular setup/platform is not needed as the benefits of coordinated ensembles are established mathematically. The revised version will explain this (see answer to question 1).
>
> ## Question 4
>
> The revised version will include the missing periods in displayed equations and correct applications of \citet/p. Thank you!

---

> > ### Author Response · Authors · 2024-11-16
> > **Uploaded revised version**
> >
> > Thank you again for the review! We uploaded a revised version that addressed all comments and increased accessibility.
> >
> > As you suggested, we included a citation and a review of [2] in which we explained why the aggregation method proposed there (a variant of independent ensembles) is inferior to ensemble coordination. This is established mathematically, and does not need experiments. Our contribution is an aggregation method that can be plugged in systems that use PATE for sequential text generation like the one in [2].
> >
> > Additionally, we expanded sections 3 and 4 to highlight our contributions and privacy analysis. As a result, the main text now exceeds the page limit, but we plan to address it by moving some of section 4.1 (proofs) to the Appendix. For now, we left it in place to facilitate an easier comparison of the versions.
> >
> > Please let us know if you have further questions. Especially on why an empirical comparison on the setup of [2] is not needed and why the advantages of our work follow from the math.

---

> > > ### Comment · Reviewer_QVdG · 2024-11-24
> > >
> > > Thank you for the detailed response. After carefully reading the rebuttal and the comments from other reviewers, I keep my original score unchanged.

---

> ### Author Response · Authors · 2024-11-26
>
> We believe we addressed your primary concerns and those of other reviewers as well, so we are puzzled by your response.
>
> Your primary concerns were
> 1. Relation to literature and review of prior work
> 2. Empirical comparison on the datasets used in some prior/concurrent works like [2]
>
>  We believe both were addressed. For [2] we explained why such a comparison is irrelevant. In a nutshell:
>  -  We propose a method, not a system. [2] proposes a system.
>  - The improvement we show is established mathematically. There is always improvement that is more significant with diversity. We also included an empirical demo as an illustration (and a simulation on varied parametrized synthetic distributions in the appendix for the purpose, as a bonus, of exploring data-dependent privacy analysis) but just taking arbitrary distributions and going further is kind of pointless.
>  - Moreover, if we "plug in" ensemble coordination in the "system" of [2] it will not only lead to significant improvement when there is diversity (and [2] works on the premise that there is) but might change what appears to be compromises made in their design.
>
> Now, let us try to explain again this last point (which appears to have been missed by the reviewer).
> Consider the case that all the teacher distributions are identical (the improvement kicks in more broadly, but this is just in order to convey the argument in a very simple setting). Suppose the prompt was "Please suggest one boy's name for my newborn." A modern model would sample a name from a distribution, likely to be non uniform and with large support, with the most likely response having probability perhaps 1/50 or  1/100. (The distribution might change by the private context but we are not considering it right now, see the paper). Ideally, we would want in this case for the privacy-preserving response to be a sample from the exact same distribution.  This is exactly what a coordinated ensemble would produce and there is no privacy loss penalty if the top probability is 1/100 or 1/1000....  One token would have all votes each time, depending on the "shared" randomness and the aggregation is successful with large privacy noise.
>
> With cold PATE, the votes split up and utility drops sharply and for low enough top probability we cannot report anything. [2] seemed to address it by changing the teacher distributions to a uniform over the top $k$ tokens. For a parameter $k$.  This is undesirable already because it looses the semantics of the well tuned base model. We do not want the distribution to be uniform. We also want it to report rare names (not only the top-$k$) with some probability.  Moreover, the aggregation still pays a penalty for $k$. The privacy loss of the aggregation depends inversely on the scale of the noise, which decreases $\propto k$ (aka the diversity). So hot PATE offers a factor $k$  gain.

---

> > ### Comment · Reviewer_QVdG · 2024-11-30
> >
> > Thank you for your detailed response. I understand that your contributions are primarily theoretical, supported by an illustrative example. Based on this, I have raised my score to 6.
> >
> > I also agree with Reviewer FQD1 that showing the practical benefits of your method through empirical evaluation on datasets used in prior works like [1,2] would greatly strengthen the paper.

---

> ### Author Response · Authors · 2024-11-30
>
> Thank you for your time and for raising the score! Regarding the comparison, it appears that [1] employed curated prompts specifically designed to elicit a single dominant response, rather than enabling free-form text generation. Consequently, we expect only minimal improvement in that setting, as it inherently involves very little diversity.
>
> As for [2], as mentioned, we anticipate that replacing their approach with coordinated ensembles would result in significant improvements and yield a privacy-preserving aggregate distribution that more closely aligns with the unmodified teacher distributions. However, the evaluation in [2], based on their GitHub page, seems to have relied on a deprecated feature of OpenAI's API (logprobs) that allowed access to the top 100 probabilities—providing high diversity. Since this feature is deprecated, we can not repeat the experiment and obtain these probability distributions. Without access to their actual collections of teacher distributions for the prompts (which do not appear to be included in their GitHub repository), we are unable to perform a direct comparison with their results.
>
> We appreciate your thoughtful feedback and will continue to explore avenues to strengthen our evaluation. Thank you once again!

---

### Official Review · Reviewer_KdZL · 2024-11-03

**Soundness:** 3
**Presentation:** 2
**Contribution:** 4
**Rating:** 6
**Confidence:** 3

**Summary:**

Private Aggregation of Teacher Ensembles (PATE) was designed for classification-like tasks where each datapoint has a single ground-truth label. For “diverse" tasks such as sequential text generation, the responses might instead be distributions. But there is a tension between diversity and privacy: diversity in the responses reduces agreement among the teachers, which in turn requires a smaller noise scale and less privacy. This paper proposes “hot PATE” which allows for higher diversity in the responses without increasing the privacy cost.

**Strengths:**

* I think this paper has a significant contribution — via a carefully designed aggregation method, PATE can now thrive in a broader and more modern setting. Formalizing the notion of “diversity-preserving” (Definition 1) is also a helpful contribution.
* The PATE framework can now be applied to very fashionable problems such as in-context learning.

**Weaknesses:**

* The paper is not beginner-friendly and seems to assume a reader who is already very familar with DP, PATE and LLMs. In fairness, this probably is going to be the chief audience of this paper, but at the same time I find it somewhat egregious that differential privacy is never formally defined (even if the definition has to be deferred to the appendix due to space constraints).
* I felt that the privacy guarantees are not rigorously stated, DP implementations are largely left as poorly-described black boxes (e.g., NoisyArgMax in Algorithm 2 is never formally introduced) and none of the algorithms include the privacy parameters as input. I didn't see a formal privacy analysis that can be easily verified, and in terms of reproducibility I feel like the algorithms can’t really be implemented without knowing, for example, how to calibrate the noise scale.

**Questions:**

* Besides coverage and diversity, are there other metrics which could be used to demonstrate the effectiveness of hot PATE?
* Line 274: If I’ve understood correctly, “the noise scale must satisfy $\sigma << \arg \max_j c_j$” is a requirement on the utility, and not the privacy? It might be helpful to explain this more thoroughly.

---

> ### Author Response · Authors · 2024-11-14
> **Re rigorous privacy guarantees**
>
> Thank you for your review! We will respond to the questions in multiple comments.
>
>
>
> ### Addressing the stated "weakness" on rigorous privacy guarantees:
> “I felt that the privacy guarantees are not rigorously stated, DP implementations are largely left as poorly-described black boxes (e.g., `NoisyArgMax' in Algorithm 2 is never formally introduced) and none of the algorithms include the privacy parameters as input. I didn't see a formal privacy analysis that can be easily verified, and in terms of reproducibility I feel like the algorithms can’t really be implemented without knowing, for example, how to calibrate the noise scale.”
>
> ### Response:
> The detailed privacy analysis can be found in Appendix D and Appendix E. While this analysis is deferred to the Appendix due to its technical nature, it primarily reflects the fact that the aggregation methods—specifically, transforming a vote histogram into a single token via NoisyArgMax or private sampling—are **standard techniques in the literature.** In particular, for NoisyArgMax, one can simply **plug in** the procedures proposed in the original PATE paper (Laplace noise) or follow up work (Gaussian noise) and apply it with the histograms produced with Hot PATE.  In the appendix, we study the tradeoffs obtained using the TCT framework (an extension of the sparse vector technique, but this is a “bonus” study that is not necessary for Hot PATE implementation).
> The key insight here is that with all these (standard) aggregation methods, the level of privacy noise required depends on the maximum frequency of any given token. That is, high consensus among teachers is best. With Cold PATE, consensus breaks down with diverse responses. In contrast, Hot PATE maintains high agreement even with diverse teacher responses—a property we establish both mathematically and empirically.
>
> In the main text, we aimed for an accessible overview that is focused on the novel components of our work. The privacy loss improvement is demonstrated via the impact on maximum token frequency, that with all aggregation methods, is directly related to the privacy loss. We demonstrate a very significant **order of magnitude** improvement. Mathematically, Hot PATE aggregation can only result in improvement, and this increases with diversity.

---

> ### Author Response · Authors · 2024-11-14
> **Response to the reviewer's questions**
>
> ### Question 1:
> “Besides coverage and diversity, are there other metrics which could be used to demonstrate the effectiveness of hot PATE?”
>
> ### Response:
> We did consider other metrics such as the TV distance between the average and “filtered” distribution (after removing counts below threshold) for independent (“cold” PATE) and coordinated (“hot” PATE) teacher ensembles (Figure 5). We also considered diversity per coverage. See also “robust average” (Remark 3 in Appendix B.2) and Figure 7. This focuses on the fraction that we can transfer “privately” versus the potential for what can be transferred privately (the robust average distribution).
>
> But overall these are the "right" metrics for our goal of transferring the diversity present in the teacher distributions.
> All our metrics demonstrate very significant gains from the hot PATE method (coordinated teacher ensembles).
>
> ### Question 2
> “Line 274”
>
> ### Response:
> Yes exactly. Often (such as in the original PATE works [Papernot et al 2018]  the noise scale $\sigma$ is scaled to the level of obtaining utility and this determines the privacy cost. (Other privacy analysis frameworks such as TCT do this as well) So the privacy loss from transferring is  inversely related to $\max_j c_j$  (the $\arg$ there is a typo which we will fix)

---

> ### Author Response · Authors · 2024-11-14
> **Beginner-friendliness**
>
> ### Reviewer's concern:
> "The paper is not beginner-friendly and seems to assume a reader who is already very familar with DP, PATE and LLMs. In fairness, this probably is going to be the chief audience of this paper, but at the same time I find it somewhat egregious that differential privacy is never formally defined (even if the definition has to be deferred to the appendix due to space constraints)."
>
> ### Response
>
> We appreciate the feedback and understand the need to make the paper more accessible. As the reviewer noted, there is a natural tradeoff in balancing technical depth with approachability for less familiar readers. The presentation in the main body is already an attempt to present concepts at a higher level.
>
> As for the absent definition of differential privacy. We agree that we should include it in the appendix, especially with parts of our considerations focused on $(\varepsilon,\delta)$-DP. It’s worth noting that our contributions and improvements are applicable across DP models (different divergences) and apply with $(\varepsilon,\delta)$ privacy but also with concentrated differential privacy (CDP and zCDP), and Rényi differential privacy (RDP). Moreover, it also applies when the goal is not privacy but robustness to “outlier" examples in the training data. This flexibility is why we initially chose to avoid a precise definition in the main text and consider metrics that transcend a particular definition.

---

> ### Author Response · Authors · 2024-11-16
> **Uploaded revised version**
>
> Thank you for your comments!  We uploaded a revised version accordingly.
>
> We made an effort to increase accessibility and implemented the following:
>
> We reorganized and expanded sections 3 and 4 in the main text. Section 3 presents our formalization of an aggregation of distribution that preserves diversity.  Section 4 contains our proposed coordinated ensembles and breaks down the presentation by including a separate subsection on establishing the privacy properties.
> We added Theorem 1 as a formal statement that connects the properties of coordinated ensembles in the lemmas in section 4 to Definition 1. The Theorem states that it suffices to only take high-count tokens in the coordinated histograms to preserve diversity.
>
> We added a separate subsection in Section 4 on the privacy properties. In particular, we added “observation 1” and “corollary 1” that highlight the fact that the privacy (sensitivity to a change in a single user) properties of the histograms generated by coordinated ensembles are identical to those generated by independent ensembles “cold PATE”. We highlighted that the gain of coordinated ensembles is due to the shape of the histograms, since high-count tokens can be reported with a lower privacy cost.
>
> We added a definition of $(\varepsilon,\delta)$-differential privacy before the analysis in Appendix D and E that explicitly uses this divergence in the TCT framework (an extension of sparse vector technique with which we evaluate data dependent analysis of composition cost). As explained, our main contribution and the established benefits of our proposed method are not specific to a particular privacy definition.
>
> The main text now exceeds the page limit, but we plan to address it by moving some of section 4.1 (proofs) to the Appendix but we left it in place for now to facilitate an easier comparison of the versions.
>
> Please let us know if the revision addresses your concerns and if you have additional questions.

---

> > ### Comment · Reviewer_KdZL · 2024-11-27
> >
> > Thanks for the response! Many of my concerns have been addressed — particularly with regards to the updated privacy analysis. I also really appreciate the efforts to make the paper more accessible (and glad to have inadvertently helped catch a typo). I’ve read the other reviews and responses and am raising my score to 6.
> >
> > While I do highly value the novelty and technical contribution of the methods proposed in this paper, I would strongly caution the authors against gatekeeping this work for an “experts-only” audience. Granted, it is inevitable that understanding many of the technical details would require a certain expertise, but ideally even a non-expert (or an expert with time constraints) could come away from this paper feeling inspired by the high-level ideas. I am sure that many readers with a similar background to reviewer FQD1 will be interested in this work, and in this case I don’t think it’s quite fair to shift the responsibility of having misunderstandings onto the reader, without first reflecting on how to improve the paper’s communication. For what it’s worth, I’m on board with reviewer FQD1’s suggestions and think that they could help the paper reach a broader audience and also clarify the practical implications of this work.

---

> > > ### Author Response · Authors · 2024-11-27
> > >
> > > Thank you for raising your score and for your thoughtful feedback! We deeply appreciate your recognition of the novelty and technical contributions of our work, as well as your constructive suggestions.
> > >
> > > Reaching a broader audience is indeed a priority for us, and gatekeeping was never our intention. We are always eager to improve our communication. While we do not know the specific background or perspective of Reviewer FQD1, we are open to further suggestions on how we can better convey the high-level ideas to inspire and engage a wider readership.
> > >
> > > If you have any additional recommendations on how to make the paper more accessible or clarify its practical implications, we would be delighted to implement them. Thank you again for your valuable input and encouragement!

---

### Official Review · Reviewer_YCGG · 2024-11-04

**Soundness:** 3
**Presentation:** 4
**Contribution:** 3
**Rating:** 8
**Confidence:** 3

**Summary:**

This paper introduces Hot PATE, an extension of the PATE (Private Aggregation of Teacher Ensembles) framework, to settings where output diversity is important. PATE works by partitioning the data and training a teacher model on each partition. Then, for a given model input, the each teacher model "votes" on a label, and a final label is privately sampled from the teacher histogram.

The key idea of Hot PATE is to preserve both privacy in the output label and the diversity of teacher distributions. The paper introduces the property of diversity preserving aggregation and introduces ensemble coordination as a technique to satisfy the property. Ensemble coordination strategically introduces correlation between teacher votes to ensure that rare tokens are transferred with high privacy noise, effectively mitigating the privacy penalty associated with high diversity, due to private sampling.

The authors provide an empirical demonstration of this approach in the context of in-context learning and show that Hot PATE yields greater diversity in output tokens.

**Strengths:**

- Introduces an extension of PATE that overcomes the diversity-privacy tradeoff
- Motivates the analysis through the notion of diversity preserving aggregation
- Connects proposed method with existing statistics literature: coordinated sampling
- Paper reads well, particularly with comparisons between hot and cold PATE

**Weaknesses:**

- The empirical analysis is more along the lines of a proof-of-concept rather than a thorough comparison. The paper would benefit from more systematic experiments between hot and cold PATE.
- No discussion of the limitations of the proposed methods. See question 1.

**Questions:**

1. In practice, does increasing diversity ever harm utility?

Other Notes:

- Typo on Line 93: "...include component that..."

- Typo on Line 190: "...two use scenarios of applications..."

- Typo on Line 323: "A tokens j that..."

---

> ### Author Response · Authors · 2024-11-13
> **Diversity and utility**
>
> Thank you for the review. As for the question: "In practice, does increasing diversity ever harm utility?"
>
> In practice, the temperature parameter in language generation models is tuned to balance the tradeoff between diversity and accuracy. Lower temperatures limit diversity, which often helps with focused factual responses but also tends to reduce creativity. Note that the tuning in current large models allow for significant diversity -- 2-7 bits of entropy per token.
>
> However, this discussion is beyond the scope of our work! Our goal is to preserve the diversity that is already present in the teacher models, which are already tuned to have the "right level" of diversity.  Methods that incur privacy-diversity tradeoff, like cold PATE, often must suppress diversity for utility. Hot PATE allows for diversity to be transferred with no privacy loss by producing correlated vote histograms via shared randomness.
>
> For instance, consider the prompt “I like my ***” in a conversation about pets that follows examples in a private context (say each teacher sees data of 10 different families from a town that has only dogs, cats, or frogs for pets). In this case we do want the model to randomly choose from these 3 good answers. Suppose most teacher models assign a probability of  ⅓ to each, as expected. Now, with Cold PATE, the histogram would have 1/3 of teachers' votes for each pet type. With hot PATE, the histogram would have nearly all votes for the same option, depending on the shared randomness, also with  probability 1/3 to each.  This means that with hot PATE it suffices to use X3 the noise scale than with cold  PATE.  Again, this diversity is something that is present in an already-tuned model. It is not something we add. Hot PATE just allows for better utility-privacy tradeoff when transferring it.

---

> ### Author Response · Authors · 2024-11-16
> **A revised version is uploaded**
>
> Thank you for your comments!  We uploaded a revised version accordingly and implemented your suggestions.
>
> We made an effort to increase accessibility and reorganized and expanded section 4 in the main text to highlight the privacy properties of coordinated ensembles.
> The main text now exceeds the page limit, but we plan to address it by moving some of section 4.1 (proofs) to the Appendix but we left it in place for now to facilitate an easier comparison of the versions.
>
> As for your comment on additional empirical evaluation: The benefits of our method -- coordinated ensembles -- is established mathematically. We gain whenever there is diversity and we know that there is diversity.  We believe that a proof of concept demonstration is sufficient for this purpose.
>
> As for your question on limitations:  In terms of privacy and utility, coordinated ensembles are always more favorable than independent ensembles. This is established mathematically. But as we mention in Section 4.3, for the in-context learning application, without API support in the LLM there would be impact on efficiency.
>
> Please let us know if the revision and our response addressed your concerns and if you have additional questions.

---

> > ### Comment · Reviewer_YCGG · 2024-11-25
> >
> > Thank you for updating the submission! This makes this phase of the reviewing process easier.
> >
> > I missed that the definition of approx. DP was missing (as reviewer Kdzl pointed out) but agree that that was an important inclusion for completeness.
> >
> > You addressed my concerns. Nice paper. I'm increasing my score to the next rung.

---

> > > ### Author Response · Authors · 2024-11-26
> > >
> > > Thank you for going through our response! You stated that you will increase your score, but it appears to still be "6"? thanks again.

---

> > > > ### Comment · Reviewer_YCGG · 2024-11-26
> > > >
> > > > Updated now

---

> > > > > ### Author Response · Authors · 2024-11-26
> > > > >
> > > > > Thank you!

---

### Official Review · Reviewer_FQD1 · 2024-11-10

**Soundness:** 2
**Presentation:** 1
**Contribution:** 3
**Rating:** 6
**Confidence:** 2

**Summary:**

The paper introduces HotPATE, a method based on the Private Aggregation of Teacher Ensemble with the distinction that the method forgoes independent teacher data and models. In fact, the teacher coordinate their sampling such that upon aggregation of their votes, rare teacher decisions (for instance, rare tokens in the case of private synthetic next-token generation) can still be produced without requiring a lot of agreement between teachers. The paper claims this process improves the diversity of the resulting vote histograms without privacy cost of not having high agreement  (which is traditionally what ensures low PATE privacy costs for private prediction). A new definition for diversity-preserving aggregeation of distributions is presented. Empirical results show that under that definition, HotPATE improves upon ColdPATE. However, practical implications of the definition and broader contribution is unclear.

**Strengths:**

- Improving diversity of PATE responses is an interesting goal, given how much the privacy of PATE comes from teacher agreement (therefore lack of diversity in teacher votes).
- The idea of coordinated sampling of tokens seems novel. Although its privacy implications are unclear.

**Weaknesses:**

As someone who is quite familiar with PATE and its derivatives, I found this paper very hard to read and digest. I think there are a couple of reasons for this:

- **A robust privacy analysis is missing.** The paper introduces a particular histogram aggregation strategy that produces rate token frequency. In a sense, this is not an aggregation that produces a single vote but rather a transformed histogram. Overall, I found the presentation of this rather simple idea overly complicated in Section 3. However, the key issue here is not the contrived procedure and Definition 1, but rather the complete lack of privacy analysis under this new aggregation method. Let me clarify this point: the PATE privacy analysis only holds under the noisy argmax release. In particualr, the analysis is a function of the gap between the top vote and the second top vote of the histogram. If we were to use Def.1 and instead release transformed vote count (for the purposes of diversity), we are strictly releasing more information. In fact, since the rare token frequencies are kept (for diversity purposes), such a scheme will likely have higher privacy cost than releasing a full noised histogram of votes.

- **Writing and exposition is not polished.** The introduction is too long and full of technical detail with frequent forward references. None of the technical terms first appearance receive proper introduction.  I find page 4 almost completely incomprehensible as a result. New terms are frequently used before they are properly defined. For instance, "homogeneous ensembles" is used in Line 186 but partially defined in Line 191. Some terms are really never properly defined at all in the introduction ("diversity", "robustness parameter", etc.)

- **Experimental results are limited.** The results are mostly validating that the algorithm produces more "diverse" tokens. I think this is necessary and good. However, throughout the paper it is unclear what the value of this "diversity" is. I was hoping the experimental results would showcases a concrete benefit from having more diverse tokens. For instance, better generalization (test error) on a down-stream task.

- **Empirical results contain no privacy quantification.** Although the paper seeks to find the trade-off between privacy and diversity, the empirical section contains no quantification of the privacy budget of the algorithm. Coupled with the fact that a proper privacy analysis is missing (see first point above) I have serious doubts regarding the privacy claims of the paper and the empirical section did not do much to alleviate them.

**Questions:**

- Can you ground your notion of diversity in a practical example? Why should one adopt your notion of diversity? What utility does it bring? Can you provide concrete empirical results to support the benefit of improved diversity as you define it?
- I had a lot of trouble with your presentation of the suggested method as a privacy-preserving algorithm. Having read the paper, I am not convinced of claims such as Line 337:
  > This high agreement allows rare tokens to pass even with high privacy noise and allow for the aggregate distribution, with fixed privacy requirements, to satisfy Definition 1.
Can you make a clear case for this?
- Have I misunderstood part of your work? To be clear, I think as is, this paper is not ready for publication. However, I want to be fair and make sure that I have not misunderstood your work. So I'll be happy to engage with you during the rebuttal process.

**Details Of Ethics Concerns:**

- Can you ground your notion of diversity in a practical example? Why should one adopt your notion of diversity? What utility does it bring? Can you provide concrete empirical results to support the benefit of improved diversity as you define it?
- I had a lot of trouble with your presentation of the suggested method as a privacy-preserving algorithm. Having read the paper, I am not convinced of claims such as Line 337:
  > This high agreement allows rare tokens to pass even with high privacy noise and allow for the aggregate distribution, with fixed privacy requirements, to satisfy Definition 1.
Can you make a clear case for this?
- Have I misunderstood part of your work? To be clear, I think as is, this paper is not ready for publication. However, I want to be fair and make sure that I have not misunderstood your work. So I'll be happy to engage with you during the rebuttal process.

---

> ### Author Response · Authors · 2024-11-14
> **Reviewer expertise mismatch? part 1 of response**
>
> Thank you for your time and feedback. We recognize that our paper presents a specialized technical contribution, which may require relevant background to fully appreciate. Regarding your last question, "Have I misunderstood part of your work,"  the review does reflect a misunderstanding of key aspects of our approach—more than just a “part” of the work. Given this, the confidence score of "3" seems higher than warranted by the level of understanding reflected in the review.
>
> Below (see multiple responses) we specifically address the claims made in the review, point to the misunderstandings, and answer the questions asked.
>
> ## "Weakness" 1:
> “ A robust privacy analysis is missing. The paper introduces a particular histogram aggregation strategy that produces rate token frequency. In a sense, this is not an aggregation that produces a single vote but rather a transformed histogram. Overall, I found the presentation of this rather simple idea overly complicated in Section 3. However, the key issue here is not the contrived procedure and Definition 1, but rather the complete lack of privacy analysis under this new aggregation method. Let me clarify this point: the PATE privacy analysis only holds under the noisy argmax release. In particualr, the analysis is a function of the gap between the top vote and the second top vote of the histogram. If we were to use Def.1 and instead release transformed vote count (for the purposes of diversity), we are strictly releasing more information. In fact, since the rare token frequencies are kept (for diversity purposes), such a scheme will likely have higher privacy cost than releasing a full noised histogram of votes.”
>
> ### Response
>
> The first major misunderstanding here is that we do not release the histogram. We release a single token each time -- just like “cold” (standard) PATE. This is explained in Section 2 that introduces the framework for Hot PATE  and also illustrated in Figure 4. We are happy to get constructive suggestions on what led to this misunderstanding of our text and figure.
>
> The heart of our method is the way the histogram is sampled (with correlated teacher votes) in Algorithm 1. You dismiss it as “contrived,” which is a poor choice of an adjective.
>
> Definition 1 is not about privacy at all. It is a formal definition of what it means for an aggregate distribution to transfer the diversity of a collection of distributions. This is our "utility." We then propose a way to do this in a privacy preserving way (via ensemble coordination). The aggregate distribution corresponds to that of the output of the probabilistic end-to-end aggregation process that produces a single token.
>
> You claim that a “robust privacy analysis is missing.” In fact, since what we change is the way the histogram is produced, and each teacher contributes a single vote, we can simply plug in the privacy analysis of cold PATE. We do state this clearly (see lines 185-187  – the fifth paragraph of page 4). Additionally, we consider in Appendix D and E additional regimes (heterogeneous ensembles) and additional privacy analysis methods (based on TCT) but the high order bit is that you can simply apply standard PATE aggregation.  Again, the primary benefit of Hot PATE, at an intuitive level,  is that the histograms are much more favorable, since there tends to be a very high count to a single token even with very diverse distributions. This is all made very precise mathematically.
>
> As for “the gap”, in fact, with PATE, if there are multiple “similar” tokens we are ok with releasing either one. Typically the parameters are not set as to separate the highest and second highest when the difference is small, especially with diversity, but so that we can release one of top votes and not release tokens with no votes or very low vote counts.
>
> We are not sure we addressed all the misunderstandings in "Weakness 1", but please read our response  and we are happy to clarify further.

---

> ### Author Response · Authors · 2024-11-14
> **Part 2 of response**
>
> ## "Weakness" 2:
>
> “Writing and exposition is not polished. The introduction is too long and full of technical detail with frequent forward references. None of the technical terms first appearance receive proper introduction. I find page 4 almost completely incomprehensible as a result. New terms are frequently used before they are properly defined. For instance, "homogeneous ensembles" is used in Line 186 but partially defined in Line 191. Some terms are really never properly defined at all in the introduction ("diversity", "robustness parameter", etc.)”
>
> ### Response
>
> Our work is a technical contribution that is established formally. It must have technical details and we deferred what we could to the appendix. There are forward references to more details in order to make the presentation more accessible, which is standard practice. If you have constructive suggestions, we are happy to implement them.  As for “homogeneous” ensembles in line 186, this is the “working assumption” of “sample and aggregate” DP  schemes like  PATE. We are happy not to use that term in line 186. You claim that the  “robustness parameter” $\tau$ is not formally defined, but this is incorrect. In the first mention (page 3) its purpose is explained at a higher level. We state it is a parameter and then state how it is used. We then fully formalize this in Definition 1 (That the reviewer indicated somehow they had read (see "Weakness" 1) but apparently missed that part?). Finally, you claim that we do not define "diversity." We say exactly what we refer to -- “diversity” is simply the notion that there are “multiple good answers” which is the reality with generative tasks.  The conditional distribution of the next token and the fact that it is not simply point mass means there is diversity.  We do formalize the notion of  “transferring diversity”.
>
> ## "Weakness" 3:
>
> "Experimental results are limited. The results are mostly validating that the algorithm produces more "diverse" tokens. I think this is necessary and good. However, throughout the paper it is unclear what the value of this "diversity" is. I was hoping the experimental results would showcases a concrete benefit from having more diverse tokens. For instance, better generalization (test error) on a down-stream task."
>
> ### Response
>
> The misunderstanding by the reviewer here is that the diversity is not something we generate but something that is **present** in the teacher’s distributions. A teacher distribution is simply that of a tuned LLMl that is applied to the sensitive data. Our method is about allowing this diversity to be transferred in a privacy preserving way to the final privacy-preserving distribution. This is exactly what we demonstrate experimentally. Essentially, for a given privacy loss, our privacy-preserving token distribution is much closer to the average teacher distribution than what is produced by cold  PATE aggregation. There is an order of magnitude improvement. Therefore, in short, we are not evaluating the LLM (llama 8B in this case). We take it as a given and there is no actual need to validate “generalization.” The goal is simply to transfer more effectively what it already does and not lose it in our privacy-preserving mechanism.
>
> ## "Weakness" 4:
>
> "Empirical results contain no privacy quantification. Although the paper seeks to find the trade-off between privacy and diversity, the empirical section contains no quantification of the privacy budget of the algorithm. Coupled with the fact that a proper privacy analysis is missing (see first point above) I have serious doubts regarding the privacy claims of the paper and the empirical section did not do much to alleviate them."
>
> ### Response:
>
> What we compare is the baseline “independent ensembles” and the proposed “coordinated ensembles”. They are compared on the coverage and diversity of “filtered” histograms that filter out the counts that are below a threshold $T$. Note that the histogram produced using shared randomness (coordinated ensembles) has the same privacy properties (L1 distance 2 between neighboring datasets) as a histogram produced via "cold" PATE.  The only difference is the shape of the histograms, which is what we evaluate. The metric of coverage and diversity after filtering precisely captures what we want since the noise scale (privacy loss) determines that effective threshold. Doing the comparison this way is cleaner as it is not specific to the particular noise distribution (Laplace or Gaussian).

---

> ### Author Response · Authors · 2024-11-14
> **Part 3 of response**
>
> ## Question 1
>
> “Can you ground your notion of diversity in a practical example? Why should one adopt your notion of diversity? What utility does it bring? Can you provide concrete empirical results to support the benefit of improved diversity as you define it?
>
> ### Answer:
>
> It is not “our notion” of diversity. Diversity is present in current LLMs in that the next-token distribution is a distribution and not a point mass. LLMs are tuned (via the temperature parameter) to allow for significant diversity – often 2-7 bits of entropy per token. When you prompt an LLM, it will sample different responses.
>
> Note that our goal is not to “create” diversity but simply to allow for it to be transferred effectively in a privacy preserving way. Simply to  emulate as effectively as  we can (while preserving  privacy) what an LLM prompted with sensitive data would do. We measure performance by the effectiveness of this “transfer.”
>
> ## Question 2
>
> “I had a lot of trouble with your presentation of the suggested method as a privacy-preserving algorithm. Having read the paper, I am not convinced of claims such as Line 337:
> This high agreement allows rare tokens to pass even with high privacy noise and allow for the aggregate distribution, with fixed privacy requirements, to satisfy Definition 1.
> Can you make a clear case for this?”
>
> ### Answer:
>
> Coordinated ensembles produce histograms that have “high agreement” on  a single (or very few) tokens. Magically, this happens even with high diversity, which is exactly the point of Hot PATE.  You might want to look at Figure 10 to see an illustration. With coordinated ensembles, each time we sample the histogram the agreement token might be different.  Now,  high agreement (high count) means that we can use more noise. More noise means that the privacy loss is lower.

---

> ### Author Response · Authors · 2024-11-16
> **Revised version is uploaded**
>
> Thank you again for your review. As explained in our response, the review showed significant misunderstanding of our results. Our revised version is an attempt to increase accessibility of our work and we hope it will be helpful. Please let us know if you have further questions.

---

> > ### Comment · Reviewer_FQD1 · 2024-11-26
> > **Thank you for the rebuttal**
> >
> > I have read the rebuttal and the updated sections of the paper (in particular section 4.2, which brings back the privacy analysis) and have gone through other reviews and rebuttal responses.
> >
> > I think section 4.2 helps to clarify the privacy guarantee.  I think the paper would benefit from bringing in data-dependent privacy analysis in the appendix in the form of an informally stated theorem; given that's really where PATE shines.
> >
> > I think the rebuttal does not alleviate the rest of my concerns regarding exposition and empirical results. I see that similar concerns have been raised by other reviewers. Regarding the latter, I get the author's argument that they are seeking to implement a general algorithm that is more useful for general-purpose GenAI tasks. They take the "diversity" of the tokens generated as a proxy measure for such general utility. This might as well be true, but would it not be better to clearly demonstrate these benefits in concrete scenarios?
> >
> > Finally, regarding the writing and exposition, I want to say that I am very squarely in the audience of your paper. If I had to read your introduction 3 times then I do not believe the paper, as it stands, is ready for publication.
> >
> > As for concrete suggestions, here are mine:
> > - Significantly shorten the introduction
> > - Focus on the application scenario and demonstrate improvement, for example, over established multi-task benchmarks for GenAI
> > - Emphasize the privacy analysis
> > - Revamp the empirical section with concrete applications. Show how improved diversity helps these applications.
> > - As for theory, take a page from Papernot et. al 2018 and provide informal theorems if theory is too much for the main paper
> >
> > Given the updated privacy analysis and the rebuttal responses, I have updated my score to 6 and reduced my confidence to 2. I will not be increasing my score any further.

---

> > > ### Author Response · Authors · 2024-11-26
> > >
> > > Thank you for raising the score and lowering the confidence level—we sincerely appreciate your effort and the thoughtful suggestions. While some of your comments still reflect certain misunderstandings, we value your acknowledgment in deferring the decision to reviewers who fully grasp the contribution.
> > >
> > > Fully appreciating our contribution, particularly in the context of sequential text generation, requires familiarity with LLMs, a solid understanding of PATE and differential privacy theory, and a significant degree of mathematical background and sophistication. This level of expertise is understandably more demanding than what many researchers in empirical privacy may possess, making the review process more challenging. Our long introduction was intentionally designed to make the ideas and motivation accessible. However, without the necessary background, readers may need to pause, reflect, and re-read certain parts to grasp the ideas. The numerous papers published over the past year applying PATE with prompts and diversity—while missing the key issues we address—underscore that the ideas are far from obvious.
> > >
> > > As for the suggestions. Thank you again for your time and effort.
> > >
> > > -- "multi-task benchmark": We have included an empirical demonstration of the benefits of our method. Given that we propose an improved aggregation method that delivers measurable benefits per generated token—validated mathematically—we do not believe it is necessary to use "multi-task benchmarks for GenAI" as part of our evaluation. The focus of our work is on the aggregation method itself, rather than a broad system-level benchmark.
> > >
> > > -- Our original manuscript assumed familiarity with differential privacy, allowing us to focus on introducing novel ideas and contributions that are less familiar to privacy researchers. However, as per your and other reviewers suggestion, the revised version now includes a more explicit discussion of the privacy properties in the main text to address potential gaps in understanding.
> > >
> > > -- "Taking a Page" from Papernot et al. (2018):
> > > While we appreciate the suggestion, we see limited value in reciting Papernot (2017, 2018). We do state that their privacy-preserving aggregation techniques can be seamlessly "plugged in" to our method. This aspect is not the novelty of our work. Furthermore, we provide an alternative, data-dependent privacy analysis in Appendix E.

---

### Meta-Review · Area_Chair_W4sC · 2024-12-18

**Metareview:**

The paper proposes a transformation that can be applied before PATE to increase consistency of expert predictions in case they share a low-probability prediction. The authors suggest this allows using private aggregation for new tasks such as aggregating next-word predictions from LLMs.

**Strengths:**
* Interesting idea
* Promising initial experiments

**Weaknesses:**
* Unconventional presentation that many reviewers found inaccessible despite being experts in the area (median score for presentation 2)
* Missing concrete privacy analysis in terms of theorems
* No experimental results for an actual end task

Based on the reviews and my own reading of the paper, I very much agree with the assessment and recommendations for improvements by Reviewer FQD1.

While the idea is interesting and the initial experiments are promising, I find the paper to be too premature for publication at ICLR. I appreciate the authors' claim of generality of their approach, but do not find this an acceptable excuse for not providing concrete privacy-utility tradeoff results and privacy analysis for an actual end task, with comparisons with previous state-of-the-art. Such results would be vital for potential users of the method to understand its value - while you demonstrate improvements in an intermediate metric, there is no guarantee on whether these will actually translate to significant improvements in performance in a relevant end task.

In terms of writing of the paper, I would strongly urge the authors to take the feedback of **all** reviewers seriously. The reviewers are experts in the field and represent potential readers of your paper. If they do not understand the paper, you should not blame the reviewers but think how you could write the paper better to avoid such misunderstandings in the future.

**Additional Comments On Reviewer Discussion:**

While all reviewers have scores suggesting acceptance, in further discussion none of them was willing to champion the paper for acceptance.

The reviewers also found the authors' tone in the responses inappropriate in dismissing the expertise of the reviewers, bordering on violating the ICLR Code of Conduct.

---

### Decision · Program_Chairs · 2025-01-22

Reject